

# Relative Importance of High-Latitude Local and Long-Range Transported Dust to Arctic Ice Nucleating Particles and Impacts on Arctic Mixed-Phase Clouds

Yang Shi[1], Xiaohong Liu[1], Mingxuan Wu[2], Ziming Ke[1], and Hunter Brown[1]

[1]Department of Atmospheric Sciences, Texas A&M University, College Station, TX, USA
[2]Atmospheric Sciences and Global Change Division, Pacific Northwest National Laboratory, Richland, WA, USA

*Correspondence to*: Xiaohong Liu (xiaohong.liu@tamu.edu)



**Abstract.** Dust particles, serving as ice nucleating particles (INPs), may impact the Arctic
surface energy budget and regional climate by modulating the mixed-phase cloud properties and
lifetime. In addition to long-range transport from low latitude deserts, dust particles in the Arctic
can originate from local sources. However, the importance of high latitude dust (HLD) as a
source of Arctic INPs (compared to low latitude dust (LLD)) and its effects on Arctic mixed-
phase clouds are overlooked. In this study, we evaluate the contribution to Arctic dust loading
and INP population from HLD and six LLD source regions by implementing a source-tagging
technique for dust aerosols in version 1 of the US Department of Energy's Energy Exascale
Earth System Model (E3SMv1). Our results show that HLD is responsible for 30.7% of the total
dust burden in the Arctic, whereas LLD from Asia and North Africa contribute 44.2% and
24.2%, respectively. Due to its limited vertical transport as a result of stable boundary layers,
HLD contributes more in the lower troposphere, especially in boreal summer and autumn when
the HLD emissions are stronger. LLD from North Africa and East Asia dominates the dust
loading in the upper troposphere with peak contributions in boreal spring and winter. The
modeled INP concentrations show a better agreement with both ground and aircraft INP
measurements in the Arctic when including HLD INPs. The HLD INPs are found to induce a net
cooling effect (-0.24 W m$^{-2}$ above 60°N) on the Arctic surface downwelling radiative flux by
changing the cloud phase of the Arctic mixed-phase clouds. The magnitude of this cooling is
larger than those induced by North African and East Asian dust (0.08 and -0.06 W m$^{-2}$,
respectively), mainly due to different seasonalities of HLD and LLD. Uncertainties of this study
are discussed, which highlights the importance of further constraining the HLD emissions.



## 1 Introduction

The Arctic has experienced long-term climate changes, including rapid warming and shrinking
in sea ice extent. Arctic mixed-phase clouds (AMPCs), which occur frequently throughout the
year, strongly impact the surface and atmospheric energy budget and are one of the main
components driving the Arctic climate (Morrison et al., 2012; Shupe and Intrieri, 2004; Tan and
Storelvmo, 2019). The AMPCs lifetime, properties, and radiative effects are closely connected to
the primary ice formation process, as the formed ice crystals grow at the expense of cloud liquid
droplets through the Wegener-Bergeron-Findeisen (WBF) process (Liu et al., 2011; M. Zhang et
al., 2019). Large ice crystals with higher fall speeds than liquid droplets can readily initiate
precipitation and further deplete cloud liquid through the riming process.
Primary ice formation in mixed-phase clouds only occurs heterogeneously with the aid of ice
nucleating particles (INPs). According to Vali et al. (1985), the heterogeneous ice nucleation is
classified into four different modes: through the collision of an INP particle with supercool liquid
droplet (contact freezing), by an INP particle immersed in a liquid droplet (immersion freezing),
when the INP particle also serves as a cloud condensation nucleus (condensation freezing), or by
the direct deposition of water vapor to a dry INP particle (deposition nucleation). The immersion
freezing is usually treated together with condensation freezing in models, as instruments cannot
distinguish between them (Vali et al., 2015). This immersion/condensation freezing is generally
thought to be the most important ice nucleation mode in the mixed-phase clouds (de Boer et al.,
2011; Prenni et al., 2009; Westbrook and Illingworth, 2013). It remains a significant challenge to
characterize the INP types and concentrations, partially because only a very small fraction of
aerosols can serve as INPs (DeMott et al., 2010). This is especially the case for the clean





environment in the Arctic. Therefore, the potential sources and amounts of Arctic INPs are still
largely unknown.
Mineral dust aerosols are identified as one of the most important types of INPs in the
atmosphere due to their high ice nucleation efficiency (DeMott et al., 2003; Hoose and Möhler,
2012; Murray et al., 2012; Atkinson et al., 2013) and their abundance in the atmosphere (Kinne
et al., 2006). They are mainly emitted from arid and semi-arid regions located at low- to mid-
latitudes, such as North Africa, the Middle East, and Asia. Observational studies found that LLD
can be transported to the Arctic (Bory et al., 2003; VanCuren et al., 2012; Huang et al., 2015)
and act as a key contributor to the Arctic INP population (Si et al., 2019). A modelling study also
suggested that low latitude dust (LLD) has a large contribution to dust concentrations in the
upper troposphere of the Arctic (Groot Zwaaftink et al., 2016), since LLD is usually lifted by
convection and topography and then transported poleward following slantwise isentropes. This
finding confirms the potential of LLD to serve as INPs in AMPCs. The impact of LLD INPs on
clouds was further investigated by Shi and Liu (2019), who found that LLD INPs induce a net
cooling cloud radiative effect in the Arctic, due to their impacts on cloud water path and cloud
fraction.
Although LLD has attracted much of the attention in the past, it is recognized that 2–3% of the
global dust emission is produced by local Arctic sources above 50°N (Bullard et al., 2016),
which include Iceland (Arnalds et al., 2016; Dagsson-Waldhauserova et al., 2014; Prospero et al.,
2012), Svalbard (Dörnbrack et al., 2010), Alaska (Crusius et al., 2011), and Greenland (Bullard
and Austin, 2011). Groot Zwaaftink et al. (2016) found that high latitude dust (HLD) contributes
27% of the total dust burden in the Arctic. Different from LLD, most of the emitted HLD is





restricted at the lower altitudes in the Arctic, because of the stratified atmosphere in the cold
environment (Bullard, 2017; Groot Zwaaftink et al., 2016).

It is also noted that HLD is likely an important source to the observed INPs in the Arctic,

especially during the warm seasons. For example, Irish et al. (2019) suggested that mineral dust
from Arctic bare lands (likely eastern Greenland or north-western continental Canada) is an
important contributor to the INP population in the Canadian Arctic marine boundary layer during
summer 2014. Attempts have been made to quantify the ice nucleating ability of HLD.
Paramonov et al. (2018) found that the Icelandic glaciogenic silt had a similar ice nucleating
ability as LLD at temperatures below -30 °C. Similarly, Sanchez-Marroquin et al. (2020)
suggested that the ice nucleating ability of aircraft-collected Icelandic dust samples is slightly
lower but comparable with that of the LLD. Some other studies also noticed that HLD can act as
efficient INPs at warm temperatures. As early as the 1950s, the airborne dry dust particles from
permafrost ground at Thule, Greenland, were found to nucleate ice at temperatures as warm as -5
°C (Fenn and Weickmann, 1959). This is corroborated by a more recent study which investigated
the glacial outwash sediments in Svalbard and ascribed the remarkably high ice nucleating
ability to the presence of soil organic matter (Tobo et al., 2019).

Despite their potential importance, HLD sources are largely underestimated or even omitted in

global models (Zender et al., 2003). Fan (2013) noticed that the autumn peak in measured dust
concentrations at Alert was underestimated by the model, likely due to a lack of local dust
emission. Similarly, Shi and Liu (2019) also mentioned that the distinction of simulated and
satellite retrieved dust vertical extinction in the Arctic became larger near the surface.

In this study, we account for the HLD dust emission by replacing the default dust emission

scheme (Zender et al., 2003) with the Kok et al. (2014a, b) scheme in the Energy Exascale Earth





System Model version 1 (E3SMv1). We further track explicitly the dust aerosols emitted from
the Arctic (HLD) and six major LLD sources using a newly developed source-tagging technique
in E3SMv1. The objectives of this study are to (1) examine the source attribution of the Arctic
dust aerosols in the planetary boundary layer and in the free troposphere; (2) examine the
contribution of dust from various sources to the Arctic dust INPs; and (3) quantify the
subsequent influence of dust INPs from various sources on the Arctic mixed-phase cloud
radiative effects. We are particularly interested in the relative importance of local HLD versus
long-range transported LLD.

The paper is organized as follows. The E3SMv1 model and experiments setup are introduced

in Section 2. Section 3 presents model results and comparisons with observations. The
uncertainties are discussed in Section 4, and Section 5 summarizes the results.
**2 Methods**
**2.1 Model description and experiment setup**

Experiments in this study are performed using the atmosphere component (EAMv1) of the U.S.

Department of Energy (DOE) E3SMv1 model (Rasch et al., 2019). The model predicts number
and mass mixing ratios of seven aerosol species (i.e., mineral dust, black carbon (BC), primary
organic aerosol, secondary organic aerosol, sulfate, sea salt, and marine organic aerosol (MOA))
through a four-mode version of modal aerosol module (MAM4) (Liu et al., 2016; Wang et al.,
2020). The four aerosol modes are Aitken, accumulation, coarse, and primary-carbon modes,
while dust aerosols are carried in accumulation and coarse modes. Aerosol optical properties in
each mode is parameterized following Ghan and Zaveri (2007). The dust optics used in this study
are updated according to Albani et al. (2014).





EAMv1 includes a two-moment stratiform cloud microphysics scheme (MG2) (Gettelman and
Morrison, 2015), with the heterogenous ice nucleation in mixed-phase clouds following the
classical nucleation theory (CNT) (Y. Wang et al., 2014). Immersion/condensation, contact, and
deposition nucleation on dust and BC are treated in the CNT scheme. It should be noted that the
WBF process rate in EAMv1 is tuned down by a factor of 10, which results in more prevalent
supercooled liquid water clouds in high latitudes than observations and many other global
climate models (Y. Zhang et al., 2019; Zhang et al., 2020). In addition, the Cloud Layers Unified
By Binormals (CLUBB) parameterization (Bogenschutz et al., 2013; Golaz and Larson, 2002;
Larson et al., 2002) is used to unify the treatments of planetary boundary layer turbulence,
shallow convection, and cloud macrophysics. Deep convection is treated by the Zhang and
McFarlane (1995) scheme.
The experiments we conducted for this study is shown in Table 1. For the control experiment
(hereafter CTRL), the EAMv1 was integrated from July 2006 to the end of 2011 at 1° horizontal
resolution and 72 vertical layers. The first six months of the experiment were treated as model
spin-up and the last five-year results were used in analyses. The horizontal wind components
were nudged to the MERRA2 meteorology with a relaxation timescale of 6 hour (Zhang et al.,
2014). In addition to CTRL, we conducted three sensitivity experiments to investigate the INP
effect of dust from major source regions. In these sensitivity experiments, heterogeneous ice
nucleation in the mixed-phase clouds by dust from local Arctic sources, North Africa, and East
Asia is turned off (i.e., noArc, noNAf, and noEAs, respectively). The other settings of these
three experiments are identical to CTRL. Analyses related to the sensitivity experiments are
provided in Section 3.4.



### 2.2 Dust emission parameterization and source-tagging technique


Dust emission in the default EAMv1 is parameterized following Zender et al. (2003) (Z03),
which uses semi-empirical dust source functions to address the spatial variability in soil
erodibility. The HLD emission is omitted in the Z03 scheme, since it was thought to be dubious
(Zender et al., 2003). In this study, we replaced the Z03 scheme with another dust emission
parameterization (Kok et al., 2014a, b) (K14) that avoids using a source function. The K14
scheme is able to produce the HLD emission over Iceland, the Greenland coast, Canada,
Svalbard, and North Eurasia (Figure 1a). Furthermore, to address the overestimation in dust
emission in clay size ($< 2 \ \mu$m diameter) (Kok et al., 2017), we changed the size distribution of
emitted dust particles from Z03 to that based on the brittle fragmentation theory (Kok, 2011). 1.1%
of the total dust mass is emitted to the accumulation mode and 98.9% of that is emitted to the
coarse mode based on the brittle fragmentation theory, whereas the fractions are 3.2% and 96.8%,
respectively in Z03.
To quantify the source attribution of dust, we implemented a dust source-tagging technique in
EAMv1. This modeling tool was previously applied to BC (H. Wang et al., 2014; Yang et al.,
2017b), sulfate (Yang et al., 2017a), and primary organic aerosol (Yang et al., 2018) in the
Community Atmosphere Model version 5 (CAM5). With this method, dust originating from
different sources can be tracked explicitly in a single model experiment. As shown in Figure 1a,
dust emissions from 7 source regions are tagged: Arctic (Arc; above 60°N, HLD source), North
America (NAm), North Africa (NAf), Central Asia (CAs), Middle East and South Asia (MSA),
East Asia (EAs), and rest of the world (RoW). The Arctic source is further divided into four sub-
sources: Alaska (Ala), North Canada (NCa), Greenland and Iceland (GrI), and North Eurasia
(NEu) (Figure S1), which are used in the analysis of INP sources in Section 3.3. RoW represents





the three major dust sources in the Southern Hemisphere (South America, South Africa, and
Australia), along with very low emissions from Europe and the Antarctic.
The global dust emission for CTRL is 5640 Tg yr$^{-1}$, which is tuned so that the global average
dust aerosol optical depth (DOD) is 0.031. This is within the range of the observational estimate
(0.030±0.005) by Ridley et al. (2016). To maintain the magnitude of the global averaged DOD,
our tuned global dust emission exceeds the range of the AeroCom (Aerosol Comparisons
between Observations and Models) models (500 to 4400 Tg yr$^{-1}$; Huneeus et al., 2011), likely
due to a short lifetime caused by too strong dust dry deposition at the bottom layer near the dust
source regions in EAMv1 (Wu et al., 2020). It is also about 2000 Tg yr$^{-1}$ higher than the previous
EAMv1 studies (Shi and Liu, 2019; Wu et al., 2020), because we distribute less dust mass into
the accumulation mode and more dust mass into the coarse mode based on Kok (2011). The
HLD emission is further tuned up by 10 times so that it accounts for 2.6% (144 Tg yr$^{-1}$) of the
global dust emission (Figure 1b), which is comparable with the recent estimates of 2-3% above
50°N by Bullard et al. (2016) and of 3% above 60°N by Groot Zwaaftink et al. (2016). The
majority of global dust emission is contributed from North Africa (51.9%, 2929 Tg yr$^{-1}$) and
Asia (37.7%, 2124 Tg yr$^{-1}$), with Asian emissions composed of MSA (20.2%, 1140 Tg yr$^{-1}$), EAs
(10.9%, 613 Tg yr$^{-1}$), and CAs (6.6%, 371 Tg yr$^{-1}$). NAm has a weak dust emission of 33.4 Tg
yr$^{-1}$ that only contributes 0.6% to the global emission, while the RoW has a combined
contribution of 7.3% (410 Tg yr$^{-1}$). In addition, the seasonal variations between HLD and LLD
emissions are different - the HLD (Arctic) source is more active in late summer and autumn,
while the LLD sources (e.g., NAf, MSA, EAs) peak in spring and early summer (Figure 1c).



**3 Result**
**3.1 Model validation**
To evaluate the model performance in simulating the dust cycle, we compare the model
predictions with measured aerosol optical depth (AOD), dust surface concentrations, and dust
deposition fluxes from global observation networks (Figure 2). We select and process the level
2.0 AOD data at 40 "dust-dominated" Aerosol Robotic NETwork (AERONET; Holben et al.,
1998) stations following Kok et al. (2014b). For dust surface concentrations, we use the same
measurements at 22 sites, which Huneeus et al. (2011) used for the AeroCom comparison, and
further extend the dataset with measurements at three high latitude stations: Heimaey (Prospero
et al., 2012), Alert (Fan, 2013), and Trapper Creek (Interagency Monitoring of Protected Visual
Environments; IMPROVE). It is noted that the measurements at Trapper Creek only include dust
particles smaller than 2.5 $\mu m$ and are only compared with simulated dust concentrations at the
same size range. All other concentration measurements capture dust particles below 40 $\mu m$ and
are compared with simulated dust over the whole size range (< 10 $\mu m$). The dust deposition
fluxes dataset, which including 84 stations, is also the same as Huneeus et al. (2011). The
locations of the observation network are shown in Figure 2d, with the AOD data taken close to
source regions and the dust surface concentrations and deposition fluxes measured at relatively
remote regions. The Pearson correlation coefficient (r) are provided for each comparison.
In general, the three comparisons indicate that our CTRL simulation is capable of capturing
the global dust cycle in both near the source and remote regions. As shown in Figure 2a, the
modeled AOD is within a factor of two of the observations over most of the stations. The
correlation of the AOD comparison is 0.73, which is comparable to the best performing
simulation (r = 0.72) in Kok et al. (2014b). Our model also does a fairly good job in simulating



the dust surface concentrations (Figure 2b) and produces a correlation coefficient of 0.83. For the
three high latitude sites, the model shows moderate underestimation at Heimaey and Trapper
Creek, but large positive bias at Alert (see discussion below). The correlation coefficient for
simulated dust deposition fluxes is also within the range of the AeroCom comparisons (0.08 to
0.84) in Huneeus et al. (2011). The model results over most of the sites are within one order of
magnitude difference, except at the polar regions. In particular, the model overestimates the dust
deposition flux in Greenland (red triangles in Figure 2c and 2d) by around two orders of
magnitude, likely due to too strong local emissions simulated near the coast of Greenland (Figure
1a).

The seasonal cycle of dust surface concentrations at the three Arctic stations (Heimaey, Alert,

and Trapper Creek) are shown in Figure 3, along with the contribution from seven tagged
sources. The simulated dust concentrations at Heimaey are dominated by HLD and agree well
with the observation in late summer and autumn (Figure 3a). Its annual-averaged low bias shown
in Figure 2b mainly comes from the springtime, when Prospero et al. (2012) found the observed
dust are related to dust storms in Iceland, indicating a possible underestimation in the simulated
Icelandic dust during this time. The HLD also dominates the surface dust concentrations at Alert
(Figure 3b), leading to a significant overestimation from June to September in our simulation,
which possibly implies a high bias and wrong seasonal cycle of HLD emission over Greenland
and North Canada. The Trapper Creek station is instead dominated by LLD from East Asia and
shows an underestimation for most of the year. It is noted that we only include fine dust
(diameter < 2.5 μm) for the comparison at Trapper Creek. Larger size range is likely to be more
influenced by HLD sources. The low bias here, especially that during the autumn, can be related
to the missing of local emissions from the coast of Southern Alaska (Figure 1a) that occurs most

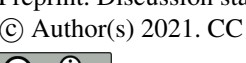



frequently in autumn (Crusius et al., 2011). An underestimation of the transport from Saharan
dust may also contribute slightly, as the influence from Saharan dust is found during mid-May at
Trapper Creek (Breider et al., 2014).
The simulated Arctic dust vertical profiles are also compared with the measured dust
concentrations during the Arctic Research of the Composition of the Troposphere from Aircraft
and Satellites (ARCTAS) flight campaign (Figure 4) (Jacob et al., 2010). The ARCTAS
campaign was conducted over the North American Arctic in April and July 2008. The simulated
profiles are averaged over the regions where the aircraft flew, in accordance with Groot
Zwaaftink et al. (2016). In April, the model does a good job in capturing the Arctic dust vertical
profiles (Figure 4a). However, in July, the model underestimates dust by a factor of 2 to 5
between 3 and 10 km (Figure 4b). It also shows an overestimation near the surface in July, which
agrees with the surface concentrations comparison at Alert station (Figure 3b). The
underestimation in the upper troposphere and overestimation near the surface likely imply a too
weak vertical transport of HLD in the North American Arctic in summertime. The high bias in
the upper troposphere may also be related to an underrepresentation of LLD transport.
Finally, we evaluate the simulated dust extinction against the Cloud-Aerosol Lidar and
Infrared Pathfinder Satellite Observation (CALIPSO) retrieval (Luo et al., 2015b, a), which
includes nighttime dust extinction for the period of 2007 to 2009. To make an apple-to-apple
comparison, the modeled dust extinction is sampled along the CALIPSO tracks and screened by
cloud fraction (Wu et al., 2020). For this comparison, we only use the first three years (2007 to
2009) of the CTRL simulation to be consistent with the observation period. Overall, the model
does a good job in capturing the Arctic dust extinction vertical profiles (Figure 5). We notice that
the simulated dust extinction is lower than CALIPSO retrievals at the upper troposphere in



summer, which agrees with the ARCTAS comparisons. Moreover, the HLD has a large
contribution in the lower troposphere in boreal summer and autumn, which is consistent with its
strong emission at that time. In contrast, LLD plays a more dominant role in the upper
troposphere, where African dust contributes the most in the springtime and East Asian dust has a
larger contribution in the other seasons.
**3.2 Arctic dust mass source attribution**
Table 2 summarizes the relative contributions from individual sources to the total Arctic dust
burden, while the transport pathways can be identified from the dust burden spatial distribution
for each source in Figure 6. We also calculate the regional burden efficiency for each source
(Table S1), which is defined as the mean contribution to the Arctic dust column burden divided
by the corresponding dust emission (H. Wang et al., 2014). This metric represents the sensitivity
of Arctic dust loading to per unit change of dust emission from each source (i.e., the poleward
transport efficiency of each source).
Our model results suggest that the HLD (Arc) is the largest contributor (30.7%) to the annual
mean Arctic dust burden among all the tagged sources. As shown in Figure 6a, the local dust is
confined within the high latitudes, with the higher amounts near the sources in North Canada,
coast of Greenland, and Iceland. The interior of the Greenland ice sheet, with its higher
elevations, is more influenced by LLD from Asia and North Africa than HLD. This is due to the
suppression of vertical transport of local emissions by the stratified atmosphere as well as limited
convection in the Arctic (Baddock et al., 2017; Bullard et al., 2016).
On the other hand, all LLD sources are responsible for 69.3% of the dust loading in the Arctic,
with considerable contributions from North Africa (24.2%) and Asia (in total 44.2%; EAs:
19.9%, MSA: 11.5%, CAs: 12.8%), and minor contributions from NAm (0.1%) and RoW (nearly





0). The North African dust is primarily transported westward to the Atlantic and southward to
Sahel, with a smaller fraction transported directly northward or northeastward across the Eurasia
to the Arctic (Figure 6c; Shao et al., 2011). The westward trajectory can also bring dust to the
Arctic through the Azores high (e.g., VauCuren et al., 2012), but this pathway is not clearly seen
on Figure 6c likely due to the strong wet removal process over the North Atlantic. As evident by
the low transport efficiency in Table S1, the significant contribution of the North African dust to
the Arctic dust burden is mainly due to its massive emission. However, this is not the case for
EAs. The East Asian dust is first lifted vertically by topography and convection (Shao et al.,
2011) and is widely spread over the Northern Hemisphere mid- and high-latitude regions through
the westerly flow in the upper troposphere (Figure 6f). The high elevation of East Asian dust
plumes results in weaker removal processes and thus an efficient poleward transport. As shown
in Table S1, the annual transport efficiency of the East Asian dust is relatively high among the
LLD sources, which is nearly three times larger than that of the North African dust. The
poleward transport of dust from CAs and MSA both takes the pathway across Siberia (Figure 6d
and 6e). The transport efficiency of the CAs dust is two times higher than that of the MSA dust
(Table S1). This is attributed to CAs being closer to the Arctic and having less southward dust
transport than MSA. The impact of NAm dust is limited by its weak emission (Figure 6b), while
dust emitted in the Southern Hemisphere (RoW) can hardly pass the equator (Figure 6g).
Earlier modeling studies (Breider et al., 2014; Groot Zwaaftink et al., 2016; Luo et al., 2003;
Tanaka and Chiba, 2006) also quantify the relative contributions of dust from various regions to
the Arctic dust loading. Among these studies, only Groot Zwaaftink et al. (2016) includes HLD.
Our estimate about the HLD percent contribution is close to that from their study (27%). For
LLD, our conclusion about the dominant role of African and Asian dust to the Arctic dust burden





is also corroborated by these previous studies. However, the relative importance of African and
Asian dust is uncertain. Based on our results, the Asian dust is responsible for 65% of the LLD
transport to the Arctic, while the African dust only contributes 35%. Other studies find that 50%
(Groot Zwaaftink et al., 2016; Luo et al., 2003; Tanaka and Chiba, 2006) to as much as 65%
(Breider et al., 2014) of the LLD in the Arctic is attributed to North Africa. These discrepancies
may be explained by the different dust emission and scavenging, dust size distribution,
meteorological fields, and/or time periods for the model simulation. For example, North Africa
dust in our study contributes slightly less (51.9%) to the global dust emission than the other
studies (from 57% to 67%). Isotopic analysis (Bory et al., 2002, 2003) and case studies (Huang
et al., 2015; Stone et al., 2005; VanCuren et al., 2012) have proved that both Asian and African
dust can be transported to the Arctic. However, it remains unclear which of them contributes
more to the Arctic dust loading due to the limited observational constraints.
HLD and LLD source regions also have very distinct vertical distributions in the Arctic.
Figures 7a and 7b show the annual mean vertical profiles of Arctic dust concentrations from
various sources and their percentage contributions, respectively. The Arctic dust in the lower
atmosphere is dominated by the local source. HLD accounts for more than 30% of the Arctic
dust concentrations below 800 hPa, with up to 85% contribution near the surface. However, due
to the weak vertical transport, the HLD contribution decreases rapidly with height and is less
than 10% above 700 hPa. In contrast, LLD has a higher contribution in the mid- and upper
troposphere than near the surface. Such a vertical distribution of LLD is consistent with Stohl
(2006) and Groot Zwaaftink et al. (2016). As Stohl (2006) found, aerosols originating from the
warm subtropics are transported poleward following the uplifted isentropes and the Arctic lower
atmosphere is dominated by the near-impenetrable cold polar dome. Therefore, there is a



slantwise lifting of low latitude aerosols during their poleward transport. NAf and EAs are the
two key contributors to the Arctic dust vertical concentrations, each of which contributes up to
one third of the total dust concentrations above 700 hPa. Dust emission from MSA also has a
moderate contribution (15-20%) that increases gradually with height, while the contribution from
CAs peaks at 700 to 800 hPa, indicating a lower altitude transport pathway than the EAs and
MSA dust.

In addition, the Arctic dust undergoes a strong seasonal cycle (Table 2 and Figure 7c-j).

Because of the strong local emissions (Figure 1c), about half of the Arctic dust burden in
summer and autumn come from HLD, with more than 50% contribution of Arctic dust
concentrations below 850 hPa in these two seasons. In contrast, LLD plays a dominant role in
spring and winter. The North African dust has the largest contribution in spring, which accounts
for about 45% of the total dust concentrations above 700 hPa. The East Asian dust is more
important in the other three seasons. Due to its high emission height, the relative contribution
from EAs tends to increase with height and reaches 30% to 50% of the total dust concentration
above 500 hPa in summer, spring, and winter.
**3.3 Immersion freezing on dust in the AMPCs**

We are particularly interested in the contribution of various dust sources to the Arctic INP

populations. Therefore, we compare the simulated INP concentrations with nine Arctic field
measurements, which are summarized in Table 3. The modeled dust INP concentrations are
diagnosed from monthly averaged aerosol properties using two ice nucleation parameterizations,
DeMott et al. (2015; hereafter as D15) and Sanchez-Marroquin et al. (2020; hereafter as SM20).
The D15 parameterization, which is derived for the Saharan and Asian dust, relates dust INP
number concentrations to the number concentration of dust particles larger than 0.5 μm diameter





and is found to produce the most reasonable LLD INP concentrations in EAMv1 (Shi and Liu,
2019). It is applied to LLD only and all the dust aerosols (LLD and HLD) in Figure 8a and
Figure 8b, respectively. The SM20 parameterization, which is derived for the HLD Icelandic
dust, describes the dust INP number concentrations as a function of surface active site density
and total dust surface area. Considering the possibly different ice nucleation ability between
HLD and LLD, we only applied the SM20 parameterization to HLD and the D15
parameterization is still applied to LLD in Figure 8c. To account for the contributions from other
aerosol types, we also calculate the INP concentrations from BC (Fig. 8d) and sea spray aerosol
(SSA; includes MOA and sea salt) (Fig. 8e) following Schill et al. (2020; hereafter as Sc20) and
McCluskey et al. (2018; hereafter as M18), respectively.
Overall, only including LLD as INPs results in up to four orders of magnitude underprediction
compared to observations (Figure 8a), while taking into account the contribution from HLD
greatly improves the model performance by increasing the simulated dust INP concentrations
(Figure 8b and 8c). The two dust parameterizations agree well with each other in simulating
HLD INPs, with SM20 producing slightly higher results than D15. Our modeling results also
indicate that BC and SSA have much less contributions to INP than dust in all the nine field
campaigns (Figure 8d and 8e).
A detailed analysis of sources of the INPs for the nine datasets based on modeling analyses
and the corresponding observations in the literature are provided in Table 3. Modeling results
indicate that HLD has larger contributions to the INPs for the campaigns conducted in summer
and autumn than spring, in agreement with the observations. Also, ground-based measurements
are more influenced by the nearby HLD sources, while LLD from EAs and NAf contributes
more to the aircraft measurements.





Our modeling analyses about the INP sources agree well with the observational studies at
Alert in spring 2016 and near Iceland in autumn 2014 (symbol "C" and "I" in Figure 8,
respectively), while the model underestimates the observed INP concentrations in both cases.
The low bias in dataset C indicates an underprediction in the long-range transport of Asian dust
to the Arctic surface in springtime. The underestimation in dataset I is more likely due to the fact
that some of the aircraft measurements were taken inside the Icelandic dust plumes (Sanchez-
Marroquin et al., 2020), which cannot be resolved by the monthly mean model output and the
coarse model horizontal resolution (1°). Such uncertainties exist in all the model-observation
comparisons.
Some other comparisons in INP sources reveal the lack of marine and carbonaceous INPs in
the model. The model results show a dominance of dust INPs in spring 2017 at Zeppelin and
Oliktok Point (symbol "D" and "E" in Figure 8) and in Autumn 2004 at Utqiaġvik (symbol "H"
in Figure 8), while the observational studies suggested the importance of marine sources at the
first two locations and of carbonaceous aerosols at Utqiaġvik. Therefore, it is likely that the
model underestimates the contribution of MOA (Wilson et al., 2015; Zhao et al., 2021a) and
does not account for terrestrial biogenic INPs (Creamean et al., 2020) due to the lack of
treatments in the model. In addition, both D15 and SM20 schemes cannot represent the high ice
nucleating ability of HLD at warm temperatures at Zeppelin in summer 2016 (symbol "G" in
Figure 8), which is attributed to soil organic matter by Tobo et al. (2019). When these organics
are taken into account in the model, model overestimation for site G will get even worse,
implying an overestimation of surface dust concentrations and/or HLD dust emission at Svalbard
in the summertime. In summary, the model's INP biases in the Arctic are likely due to biases in
the simulated aerosol fields (e.g., dust, MOA, and BC) and uncertainties in current ice nucleation



396 parameterizations or missing representations of other INP sources (e.g., terrestrial biogenic

397 aerosols).

398  The comparisons above are based on INP concentrations at a given temperature set by the INP

399 instruments, which reflects the potential INP populations under ambient aerosol conditions. Next,

400 we examine the immersion freezing rate of dust originating from the seven tagged sources

401 (Figure 9) to evaluate the influences of HLD and LLD on ice nucleation processes in mixed-

402 phase clouds. It is noted that the immersion freezing rate here is calculated online in the model

403 using the ambient temperature and the default CNT ice nucleation parameterization.

404  Compared with its contribution to the dust burdens, the contribution of the HLD to the annual

405 mean mixed-phase cloud immersion freezing rate is relatively small (~10% below 600 hPa)

406 (Figure 9a). This is because the HLD is mainly located in the lower troposphere and not a lot of

407 HLD can reach the mixed-phase cloud levels (or the freezing level), especially under the case

408 that the HLD tends to be more prevalent in the warm seasons (see more discussion below).

409 Among the LLD sources, North African dust (Figure 9c) and East Asian dust (Figure 9f) are the

410 two major contributors, both of which are responsible for more than 20% of the annual mean

411 immersion freezing rate in the mixed-phase clouds. Consistent with the vertical distribution of

412 dust concentrations, the North African dust has its maximum contribution (30-40%) at around

413 500 hPa, while the East Asian dust plays a more important role at higher altitudes (above 400

414 hPa). Dust from Central Asia also has a moderate contribution (~20%) to the immersion freezing

415 rate in the Arctic (Figure 9d).

416  Considering the different seasonality of HLD and LLD in the Arctic, we next investigate the

417 seasonal variations of the immersion freezing rate in the Arctic mixed-phase clouds from HLD

418 and two dominating LLD sources (NAf and EAs) (Figure 10). HLD has the largest contribution



to the Arctic immersion freezing rate in boreal autumn, with more than 30% below 700 hPa and
up to 50% near the surface (Figure 10c). It is related to the prevalence of HLD and relatively
cold temperatures during this time in the Arctic. This is not the case for the summer, when the
freezing level is relatively high. Although it is responsible for 50% of the total Arctic dust
burden in the boreal summer, HLD has a limited contribution to the immersion freezing rate in
the clouds (Figure 10b), because its weak vertical transport makes it hard to reach the freezing
line. The contrast results in summer and autumn suggest that the immersion freezing rate in the
Arctic clouds is influenced by air temperature in addition to the aerosols. It also implies that the
surface INP measurements may not reflect the complete picture of INP effects and more aircraft
INP measurements are needed in the future. The seasonal variations of the immersion freezing
rate from NAf and EAs are weaker than that from HLD, but are still subjected to the vertical
temperature change with season. The North African dust contributes more in spring and winter,
while the East Asian dust is more important in summer and autumn.
**3.4 Impact on cloud properties and radiative fluxes**

Dust INPs can freeze the supercooled liquid droplets, which impacts the cloud microphysical

and macrophysical properties and modulates the Earth's radiative balance. To examine such
impacts, we conduct three sensitivity experiments that turn off the heterogeneous ice nucleation
in the mixed-phase clouds by dust from Arctic local source, North Africa, and East Asia,
respectively (i.e., noArc, noNAf, and noEAs in Table 1). The impacts of dust INPs from each
source are determined by subtracting the respective sensitivity experiment from CTRL. Due to
the feedbacks in dust emission and wet scavenging caused by changing cloud properties, the dust
concentrations in the sensitivity experiments are not identical to CTRL, but the absolute
differences are mostly within 5% (Figure S2 in the supporting information).



The cloud liquid and ice changes caused by dust INPs from each source are shown in Figure
11. Due to the strengthening of heterogeneous ice nucleation processes, INPs from all the three
sources consistently reduce the total liquid mass mixing ratio (TLIQ) (Figure 11, first column)
and cloud liquid droplet number concentration (NUMLIQ) (Figure 11, third column). The
influence of HLD is mainly in the lower troposphere (Fig. 11, top row) and the influence of LLD
extends to higher altitudes (Fig. 11, bottom two rows). Moreover, the cloud ice number
concentration (NUMICE) decreases in the upper troposphere (Figure 11, fourth column), likely
due to less cloud droplets available for the homogeneous freezing in cirrus cloud after
introducing dust INPs in the mixed-phase clouds. With fewer ice crystals falling from the cirrus
clouds to the mixed-phase clouds, the WBF process in the mixed-phase clouds is inhibited
(Figure S3). Other ice phase processes such as the accretion of cloud water by snow and the
growth of ice crystals by vapor deposition also become less efficient, which decreases the total
ice mass mixing ratio (TICE) above 600-700 hPa altitude (Figure 11, second column). TICE in
the lower troposphere is increased because of immersion freezing and snow sedimentation from
above.
Since liquid water path (LWP) is found to play a critical role in the Arctic radiative budget
(e.g., Dong et al., 2010; Hofer et al., 2019; Shupe and Intrieri, 2004), we further investigate the
seasonal variations of LWP changes caused by dust INPs from the three sources (Figure 12).
Corroborated with their large contribution to the immersion freezing rate during this time (Figure
10, top row), HLD INPs produce the strongest LWP decrease (-1.3 g m$^{-2}$) in boreal autumn
(Figure 12c), especially over North Canada and Greenland. The influence of LLD INPs on LWP
peaks in spring and winter. North African dust tends to have a larger impact on North Eurasia,
while East Asian dust impacts the west Arctic more.



Dust INPs from the three sources consistently increase (decrease) the annual mean
downwelling shortwave (longwave) radiative flux (FSDS and FLDS) at the surface (Figure 13,
left and middle columns). This is mainly due to the LWP decrease, which reduces the cloud
albedo and longwave cloud emissivity. For HLD INPs, the FLDS reduction dominates over the
FSDS increase and causes a net cooling effect at the Arctic surface (-0.24 W m$^{-2}$) (Figure 13c).
In contrast, FSDS and FLDS changes related to the LLD INPs are comparable, which cancels
each other and yields a small net radiative effect (0.08 W m$^{-2}$ for NAf and -0.06 W m$^{-2}$ for EAs)
(Figure 13, bottom two rows). These differences in the net radiative effect are associated with
different seasonalities of HLD and LLD. The insolation in the Arctic is strong in spring and
summer but very limited in autumn and winter. Since the HLD INPs have much stronger
influence on LWP in autumn and winter than spring and summer (Figure 12), their contribution
to the FSDS warming is weak and the FLDS cooling in autumn and winter dominates the annual
mean effect (Table 4, part 1; also seen in Figure S4 to S6). LLD INPs are also important in
spring and summer, so their FSDS warming effect is comparable to, and compensates for, the
FLDS cooling effect.
We also examined the dust INP effect on cloud radiative forcing (CRF) at the top of the
atmosphere (TOA) (Table 4, part 2). Dust INPs from the three sources induce a small net cooling
(from -0.03 to -0.05 W m$^{-2}$) in the Arctic, with SW warming and LW cooling effects. The net
cooling persists throughout the year, except for the summertime when the sufficient insolation
results in a strong SW warming and, consequently, a net warming effect. Shi and Liu (2019) also
found LLD can induce a generally net cooling effect above 70°N (0.18 to -1.95 W m$^{-2}$), but in a
much higher magnitude than the sum of NAf and EAs dust INP effects (-0.15 W m$^{-2}$ above 70°N,



not shown in Table 4), which implies the aerosol glaciation effect on mixed-phase clouds is
highly non-linear.
**4. Discussion**
The HLD emission in our CTRL simulation is manually tuned up by 10 times to match the
estimate by Bullard et al. (2016), which is derived by compiling field measurements in Iceland
and Alaska. Since the instruments were operated under extreme Arctic conditions and the
sampling is very scarce, this estimate may have large uncertainties. Therefore, the tuned HLD
emission can be biased as well. Considering the overestimation of Greenland dust deposition,
summertime surface dust concentrations at Alert station, and surface INP concentrations at
Svalbald, our tuning may cause a regional and temporal high bias in HLD dust emissions. We
examine this uncertainty by conducting a sensitivity experiment with halving HLD emissions in
CTRL (i.e., HLD_half) and analysing the interannual variability of CTRL and HLD_half
simulations (Table S2 and Figure S7-S8). The HLD_half simulation indeed has a better
performance than CTRL. However, the high bias for Greenland deposition and the summertime
overestimation of Alert dust surface concentration still exist, which reflects the limitation of the
dust emission parameterization we use. This parameterization may not be able to capture the
spatial distribution of dust emissions across the Arctic, considering that the model performance
at other sites (e.g., Heimaey, Figure 3a) is much better. Also, the HLD emissions and their
regional distributions have large interannual variabilities. Therefore, comparing model
simulations with measurements conducted in different years may result in large uncertainties.
The overestimation of surface dust and INP concentrations may imply a too weak vertical
transport of HLD, considering the low biases of dust in the upper troposphere as compared with





ARCTAS measurements and CALIPSO retrievals. The weak vertical transport at the source
regions in EAMv1 was also found in Wu et al. (2020), which was related to the too strong dry
deposition at the surface layer. If this bias is addressed, HLD would contribute less (more) to the
Arctic dust concentrations in the lower (upper) troposphere, which suggests a larger contribution
of HLD to the heterogeneous ice nucleation in the mixed-phase clouds in the summertime. As a
result, the HLD would induce a more positive net downwelling radiative flux at the surface in
summer and a less negative annual mean radiative effect. It is also noted that the underprediction
in the upper troposphere dust may come from a weak long-range transport of LLD. If this is the
case, the HLD would have a weaker contribution to the upper level dust concentrations and
likely less of an impact on mixed-phase cloud heterogeneous ice nucleation in the summertime.
In addition, EAMv1 has intrinsic biases in its cloud microphysics parameterizations. As
mentioned in Section 2.1, the WBF process rate in EAMv1 is tuned down by a factor of 10,
which results in too many supercooled liquid clouds in high latitudes (Y. Zhang et al., 2019; M.
Zhang et al., 2020). Shi and Liu (2019) found the sign and magnitude of dust INP cloud radiative
effect in the Arctic would change, after removing the tuning factor for the WBF process in
EAMv1. Moreover, EAMv1 does not account for several secondary ice production mechanisms,
which are suggested to have a large impact on the ice crystal number concentrations and thus
cloud phase (Zhao and Liu, 2021; Zhao et al., 2021b). All these uncertainties in the cloud
microphysical processes would influence our estimate of INP radiative effect and should be
addressed in future studies.



### 5. Conclusions


In this study, we investigate the source attribution of dust aerosols in the Arctic and quantify
the relative importance of Arctic local dust versus long-range transported LLD to the Arctic dust
loading and INP population. We found that HLD is responsible for 30.7% of the total dust
burden in the Arctic, whereas LLD from Asia and North Africa contributes 44.2% and 24.2%,
respectively. The vertical transport of HLD is limited due to the stable cold air in the Arctic and
thus it contributes more to the dust burden in the lower troposphere. In boreal summer and
autumn when the contribution of HLD is at a maximum because of stronger local dust emissions,
HLD is responsible for more than 30% of the Arctic dust loading below 800 hPa, but less than 10%
above 700 hPa. In contrast, LLD from North African and East Asian dust dominates the dust
burden in the free troposphere, since the poleward transport of LLD follows the uplifted
isentropes. The North African and East Asian dust accounts for about two thirds of the dust
loading above 700 hPa, with the remaining one third from other LLD sources. The North African
dust contributes more between 500 and 700 hPa, while the East Asian dust dominates in the
upper troposphere (above 400 hPa) because of its high emission heights. In addition, the North
Africa source has a larger contribution in springtime, while the other three seasons are more
influenced by the East Asian source.
Modeled dust INP concentrations are investigated following two ice nucleation
parameterizations: D15 and SM20. Compared with INP measurements, our results show that
including HLD as INPs significantly improves the model performance in simulating Arctic INP
concentrations, especially for the ground measurements and for the measurements conducted in
summer and autumn. We also examine the INP contributions from BC and SSA based on Sc20
and M18, respectively. The model suggests that both of them are only weak sources compared





with dust. We note that the model may underestimate SSA INPs and currently misses the
representation of terrestrial biological INPs. The model biases of INPs can also be due to bias in
simulating Arctic dust concentrations and/or the uncertainties in ice nucleation parameterizations.

We examine the contribution of dust from the three sources (Arctic, North Africa, and East

Asia) to the ambient immersion freezing rate in the Arctic. The contribution from HLD shows a
strong seasonal variation, with the peak contribution in boreal autumn (above 20% below 500
hPa). In summer, although HLD has strong contributions to the dust loading and INP
concentrations in the lower troposphere, its impact on the ambient immersion freezing rate is
limited due to the warm temperatures and weak vertical transport. This finding implies that
surface INP measurements may not be sufficient in representing the INP population in the Arctic
mixed-phase clouds and more measurements of INP vertical profiles are needed in the future.
North African and East Asian dust are the two major LLD contributors to the ambient immersion
freezing rate. The annual mean contribution (30-40%) from North African dust peaks at around
500 hPa, while the immersion freezing is dominated by East Asian dust (more than 40%) in the
upper troposphere (above 400 hPa).

The cloud glaciation effect of dust INPs from local Arctic sources, and North African and East

Asian sources, is further examined. It is found that INPs from all the three sources consistently
result in a reduction in TLIQ and NUMLIQ. TICE and NUMICE at higher altitude also decrease,
likely due to the weakening of homogeneous freezing in cirrus clouds. LWP reduction caused by
HLD INPs is evident in autumn and winter, while those by dust INPs from the two LLD sources
peak in spring. HLD INPs also drive a net cooling effect of -0.24 W m$^{-2}$ in the downwelling
radiative flux at the surface in the Arctic, while the net radiative effects of the two LLD INP
sources are relatively small (0.08 W m$^{-2}$ for NAf and -0.06 W m$^{-2}$ for EAs). This variation in





radiative effect reflects the seasonal difference between HLD and LLD. Our results also suggest
that all the three dust sources result in a weak negative net cloud radiative effect (-0.03 to -0.05
W m$^{-2}$) in the Arctic, which is consistent with Shi and Liu (2019).
Overall, our study shows that the Arctic local dust, which has been overlooked in previous
studies, may have large contributions to the Arctic dust loading and INP population. It can also
influence the Arctic mixed-phase cloud properties by acting as INPs. Considering the climate
impacts of local Arctic dust emissions will be important given a warming climate, where
reduction in snow coverage and more exposure of dryland in the Arctic may lead to increased
HLD emissions.
*Code availability.* The E3SM code is available on GitHub: https://github.com/E3SM-
Project/E3SM.git.
*Author contribution.* YS and XL conceived the project. YS modified the code, conducted the
simulations, and led the analyses with suggestions from XL, MW, ZK, and HB. XL supervised
the study. YS wrote the first draft of the paper. All coauthors were involved in helpful
discussions and revised the paper.
*Competing interests.* The authors declare that they have no conflict of interest.
*Acknowledgements.* The authors would like to thank Drs. Xi Zhao, Meng Zhang, and Sarah
Brooks for their comments and suggestions. Mingxuan Wu is supported by the US Department
of Energy (DOE), Office of Biological and Environmental Researsch, Earth and Environmental
System Modeling program as part of the Energy Exascale Earth System Model (E3SM) project.



The Pacific Northwest National Laboratory (PNNL) is operated for DOE by the Battelle
Memorial Institute under contract DE-AC05-76RLO1830. This research used resources of the
National Energy Research Scientific Computing Center, a DOE Office of Science User Facility
supported by the Office of Science of the U.S. Department of Energy under contract DE-AC02-
05CH11231.
*Financial support.* This research was supported by the DOE Atmospheric System Research
(ASR) Program (grants DE-SC0020510 and DE-SC0021211).

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



**Table 1.** Experiments conducted in this study.

| Experiment | Description |
|---|---|
| CTRL | Control simulation using the CNT parameterization for heterogeneous ice nucleation and Kok et al. (2014a, b) for dust emission parameterization. |
| noArc | Same as CTRL, but turn off heterogeneous ice nucleation in mixed-phase clouds by HLD. |
| noNAf | Same as CTRL, but turn off heterogeneous ice nucleation in mixed-phase clouds by North African dust. |
| noEAs | Same as CTRL, but turn off heterogeneous ice nucleation in mixed-phase clouds by East Asian dust. |






**Table 2.** Annual and seasonal mean Arctic dust burden (mg m$^{-2}$) from different sources. The
numbers in parentheses are the relative contributions (%) of each source to the total Arctic dust
burden. The total Arctic dust burden is shown in the last row.

|  | ANN | MAM | JJA | SON | DJF |
|---|---|---|---|---|---|
| Arc | 2.1 (30.7) | 0.3 (3.9) | 5.1 (50.4) | 2.5 (47.5) | 0.5 (14.6) |
| NAm | 0.1 (0.9) | 0.1 (1.3) | 0.1 (0.6) | 0.0 (0.7) | 0.0 (1.2) |
| NAf | 1.7 (24.2) | 3.7 (41.4) | 1.5 (14.4) | 0.7 (12.9) | 0.9 (26.4) |
| CAs | 0.9 (12.8) | 1.1 (12.5) | 1.3 (13.0) | 0.8 (14.7) | 0.3 (10.1) |
| MSA | 0.8 (11.5) | 1.6 (17.9) | 0.7 (7.0) | 0.3 (6.1) | 0.6 (17.4) |
| EAs | 1.4 (19.9) | 2.0 (23.0) | 1.5 (14.7) | 0.9 (18.1) | 1.0 (30.2) |
| RoW | 0.0 (0.0) | 0.0 (0.0) | 0.0 (0.0) | 0.0 (0.0) | 0.0 (0.1) |
| Total Burden (mg m$^{-2}$) | 6.9 | 8.9 | 10.2 | 5.2 | 3.3 |






**Table 3.** Summary of the nine Arctic INP measurements used for INP comparisons in Figure 8.

|   | Location | Time | Measured platform | Reference | Possible INP source mentioned in literature | INP source attribution from modeling[+] |
|---|----------|------|-------------------|-----------|---------------------------------------------|------------------------------------------|
| A | Utqiaġvik | Spring, 2008 | Aircraft | McFarquhar et al. (2011) | Metallic or composed of dust[*] | HLD (NCa) and LLD (EAs) |
| B | Alert | Spring, 2014 | Ground-based | Mason et al. (2016) | Not mentioned | LLD (EAs) |
| C | Alert | Spring, 2016 | Ground-based | Si et al. (2019) | LLD from Gobi Desert | LLD (EAs) |
| D | Zeppelin | Spring, 2017 | Ground-based | Tobo et al. (2019) | Marine organic aerosols | HLD (NEu) |
| E | Oliktok Point | Spring, 2017 | Ground-based | Creamean et al. (2018) | Dust and primary marine aerosols | LLD (mainly from EAs and some from NAf) |
| F | Alert | Summer, 2014 | Ground-based | Mason et al. (2016) | Not mentioned | HLD (NCa) |
| G | Zeppelin | Summer, 2016 | Ground-based | Tobo et al. (2019) | HLD from Svalbard or other high latitude sources[**] | HLD (NEu) |
| H | Utqiaġvik | Autumn, 2004 | Aircraft | Prenni et al. (2007) | Dust and carbonaceous particles | HLD (NCa) and LLD (EAs) |
| I | South of Iceland | Autumn, 2014 | Aircraft | Sanchez-Marroquin et al. (2020) | Icelandic dust | Dominated by HLD (GrI), little from LLD (NAf) |

[*] Carbonate, black carbon, and organic may also contribute, according to Hiranuma et al. (2013).
[**] The HLD in this campaign is reported to have remarkably high ice nucleating ability, which
may be related to the presence of organic matter.
[+] The modeling analyses include INP contribution from HLD, LLD, BC, and SSA.





**Table 4.** Arctic averaged surface downwelling radiative fluxes and TOA cloud radiative forcing
changes caused by dust INPs originated from local Arctic sources (Arc), North Africa (NAf), and
East Asia (EAs). Units are W m$^{-2}$.

| | ANN | | | MAM | | | JJA | | | SON | | | DJF | | |
|---|---|---|---|---|---|---|---|---|---|---|---|---|---|---|---|
| | SW | LW | Net | SW | LW | Net | SW | LW | Net | SW | LW | Net | SW | LW | Net |
| Part 1. INP effect on surface downwelling radiative fluxes | | | | | | | | | | | | | | | |
| Arc | 0.11 | -0.36 | -0.24 | 0.27 | -0.31 | -0.03 | 0.12 | 0 | 0.12 | 0.04 | -0.55 | -0.51 | 0.02 | -0.56 | -0.54 |
| NAf | 0.33 | -0.25 | 0.08 | 0.78 | -0.60 | 0.19 | 0.50 | 0.01 | 0.51 | 0.02 | -0.03 | -0.02 | 0.03 | -0.39 | -0.36 |
| EAs | 0.35 | -0.41 | -0.06 | 0.68 | -0.60 | 0.09 | 0.59 | 0.02 | 0.61 | 0.08 | -0.27 | -0.19 | 0.04 | -0.80 | -0.76 |
| Part 2. INP effect on TOA cloud radiative forcing | | | | | | | | | | | | | | | |
| Arc | 0.06 | -0.11 | -0.05 | 0.06 | -0.07 | -0.01 | 0.14 | -0.02 | 0.12 | 0.03 | -0.23 | -0.20 | 0.01 | -0.12 | -0.11 |
| NAf | 0.20 | -0.23 | -0.03 | 0.34 | -0.34 | 0 | 0.41 | -0.18 | 0.24 | 0.03 | -0.20 | -0.16 | 0.02 | -0.23 | -0.21 |
| EAs | 0.20 | -0.24 | -0.04 | 0.22 | -0.23 | -0.02 | 0.46 | -0.17 | 0.29 | 0.09 | -0.29 | -0.20 | 0.02 | -0.26 | -0.24 |


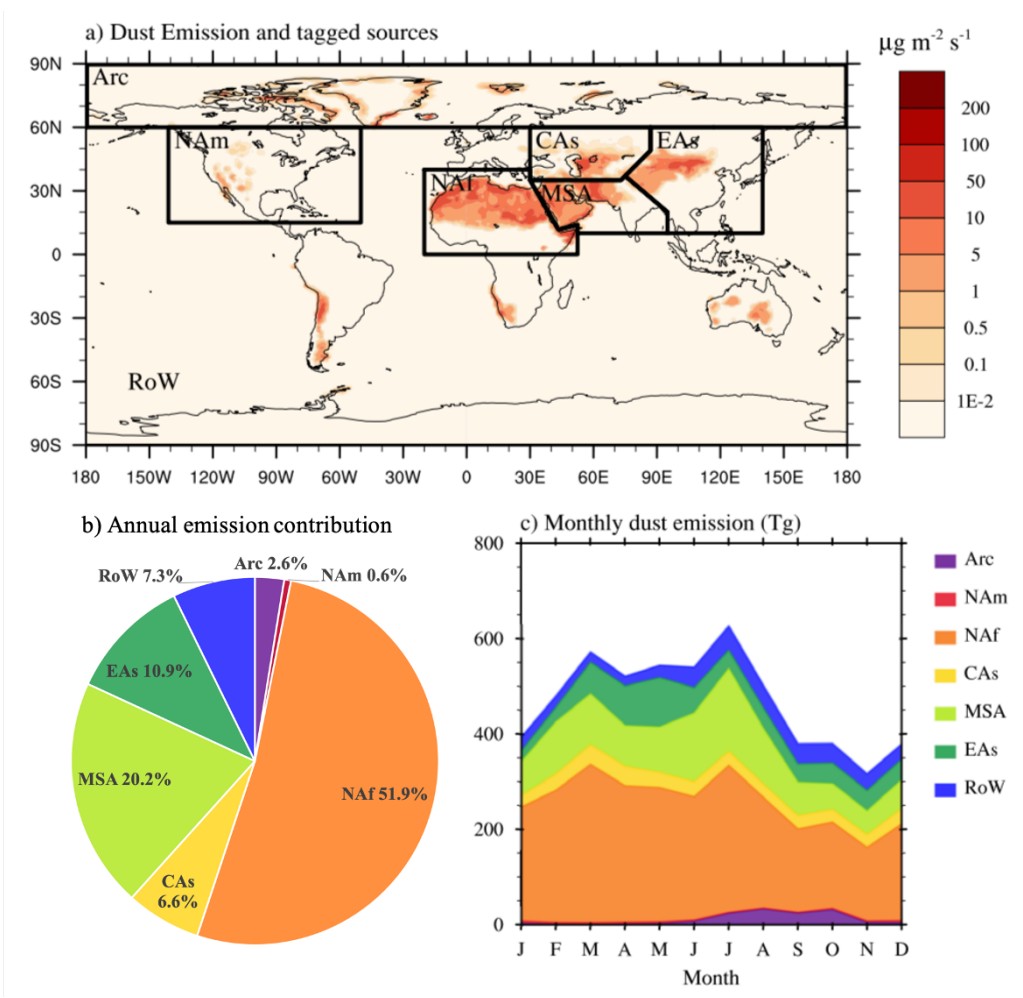


**Figure 1.** a) Simulated global annual mean dust emission with 7 tagged source regions (Arc: Arctic; NAm: North America; NAf: North Africa; CAs: Central Asia; MSA: Middle East and South Asia; EAs: East Asia; RoW: Rest of the World). b) The respective percentage contributions to the global annual mean dust emission from the individual source regions. c) Seasonal cycle of global dust emission.



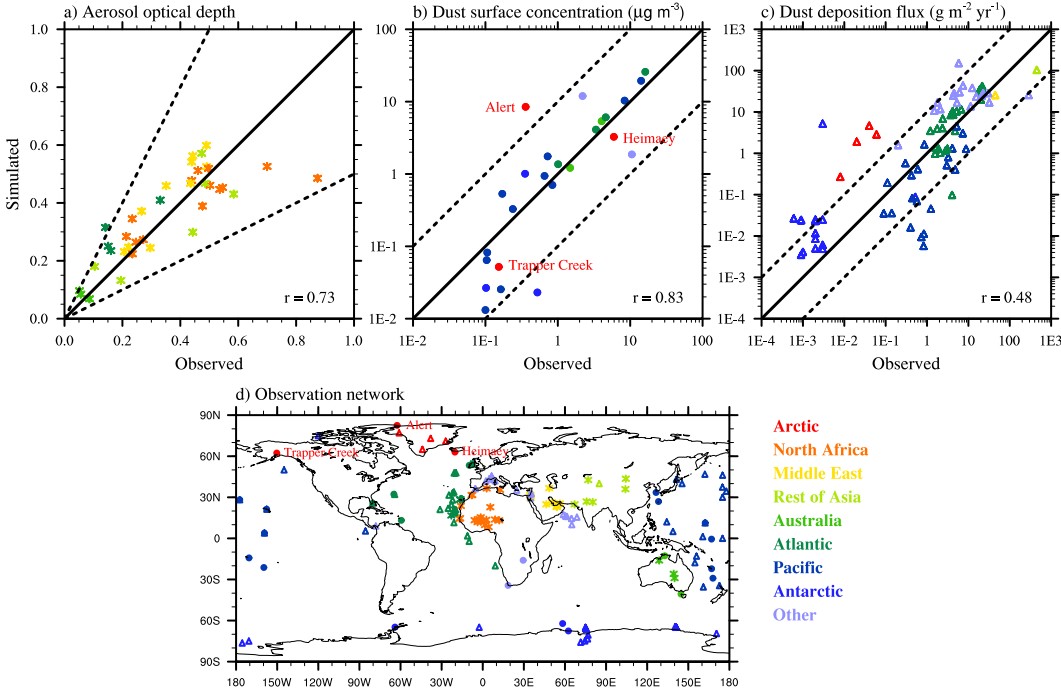

**Figure 2.** Comparison of observed and simulated a) averaged AOD at 40 dust-dominated stations (stars), b) dust surface concentration at 25 sites (circles), and c) dust deposition flux at 84 sites (triangles). Solid lines represent 1:1 comparison. Dashed lines mark 2 factor of magnitude bias in panel a) and 1 order of magnitude differences in panel b) and c). For each comparison, the correlation coefficient (r) is noted. The AOD data is conducted by AERONET. The dust surface concentration measurements include 20 stations managed by Rosenstiel School of Marine and Atmospheric Science at the University of Miami (Prospero et al., 1989; Prospero, 1996; Arimoto et al., 1995), two Australia stations (Maenhaut et al., 2000a, b), and three Arctic stations (Heimaey (Prospero et al., 2012), Alert (Fan, 2013), and Trapper Creek (IMPROVE)). The deposition fluxes data is a compilation of measurements from Ginoux et al. (2001), Mahowald et al. (2009), and the Dust Indicators and Records in Terrestrial and Marine Paleoenvironments (DIRTMAP) database (Tegen et al., 2002; Kohfeld and Harrison, 2001).





Stations are grouped regionally and classified by different colors. The locations of the

measurements are shown in panel d).

991

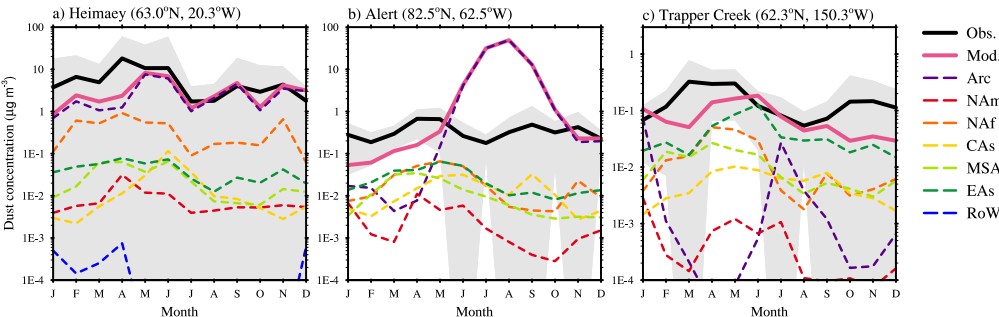

**Figure 3.** Comparison of measured (black solid line) and simulated (pink solid line) monthly mean dust surface concentration at three high latitude stations – a) Heimaey, b) Alert, and c) Trapper Creek. Contributions from seven tagged sources are shown by colored dashed lines. The locations of the three stations are shown in Figure 2d. The dust concentrations at Trapper Creek only include particles with diameter less than 2.5 μm. The other two stations include dust over the whole size range.

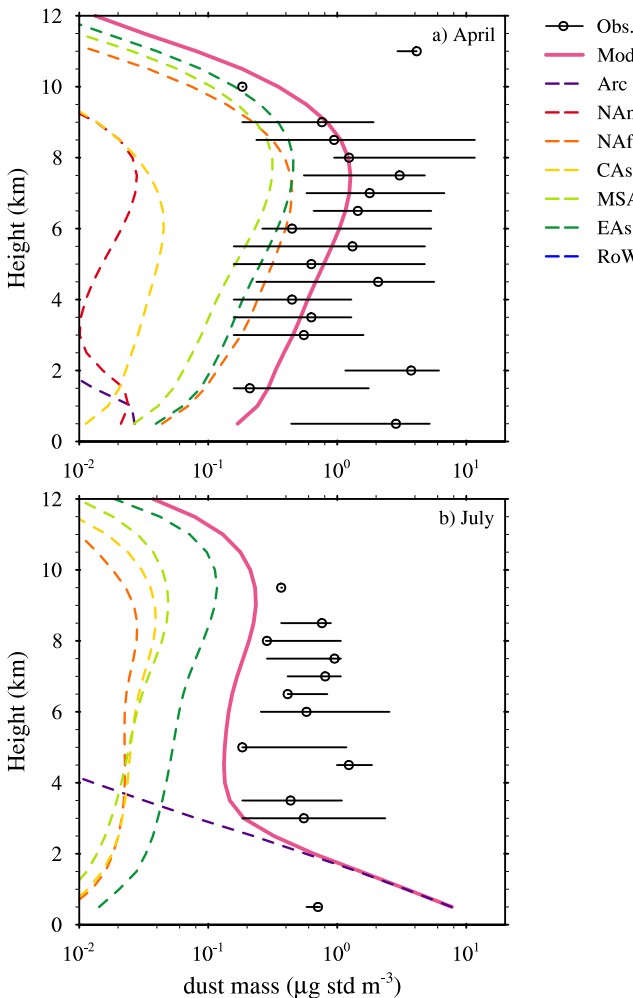

1000

**Figure 4.** Comparison of vertical dust concentrations from ARCTAS flight observations (Jacob et al., 2010) (black circle) and CTRL simulation (pink solid line) in a) April and b) July. We show median values for observations at each level. The maximum and minimum of the measurements at each level are shown by black lines. Contributions from the seven tagged sources in CTRL are shown by colored dashed lines. The ARCTAS dust mass concentrations are derived from measured calcium and sodium concentrations. The measurements data are processed using the same method as Breider et al. (2020). Briefly, we assume a calcium to dust



mass ratio of 6.8% and further correct the calcium concentrations for sea salt by assuming a
calcium to sodium ratio of 4%. Only measurements obtained north of 60°N are used for the
analyses. The low-altitude observations near Fairbanks, Barrow, and Prudhoe Bay are removed.
Also, data from below 1 km on 1, 4, 5, 9 July is removed to exclude the influence of wildfire.
The ARCTAS flight campaign was conducted in 2008, while the modeled vertical profiles are
averaged for each April and July from 2007 to 2011, respectively. Following Groot Zwaaftink et
al. (2016), the simulation profiles are averaged for the regions north of 60°N and 170°W to
35°W in April and 135°W to 35°W in July.

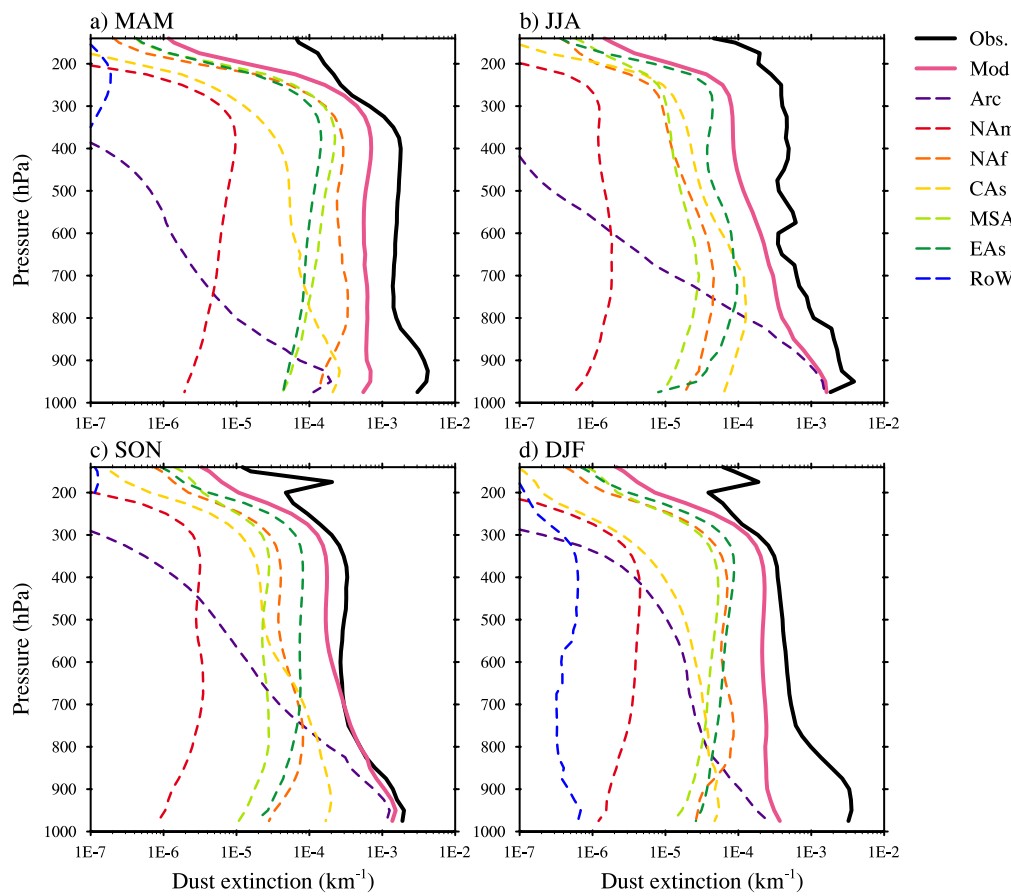


**Figure 5.** Comparison of seasonal CALIPSO retrieved (Luo et al., 2015a, b) (black solid line)

and model simulated (pink solid line) dust extinction vertical profiles in the Arctic. Contributions

from seven tagged sources are shown by colored dashed lines.



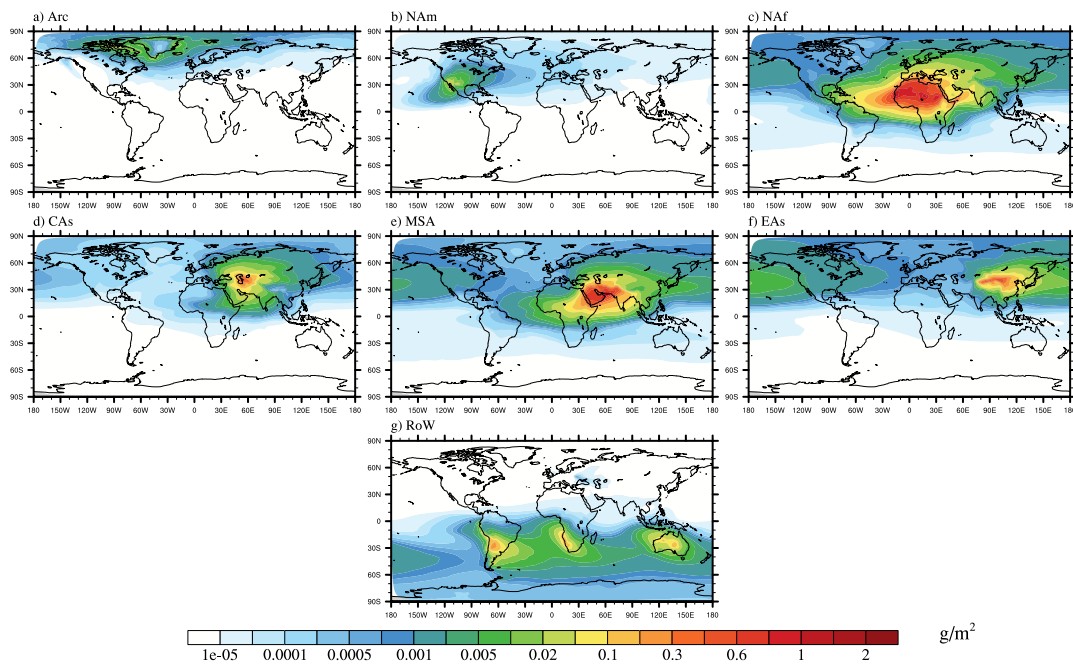


**Figure 6.** Global distribution of annual mean dust column burdens for various tagged sources.







**Figure 7.** Annual and seasonal mean Arctic vertical dust concentrations (left panel) and
percentage contributions from tagged sources (right panel). Different tagged sources are
classified by different colors.



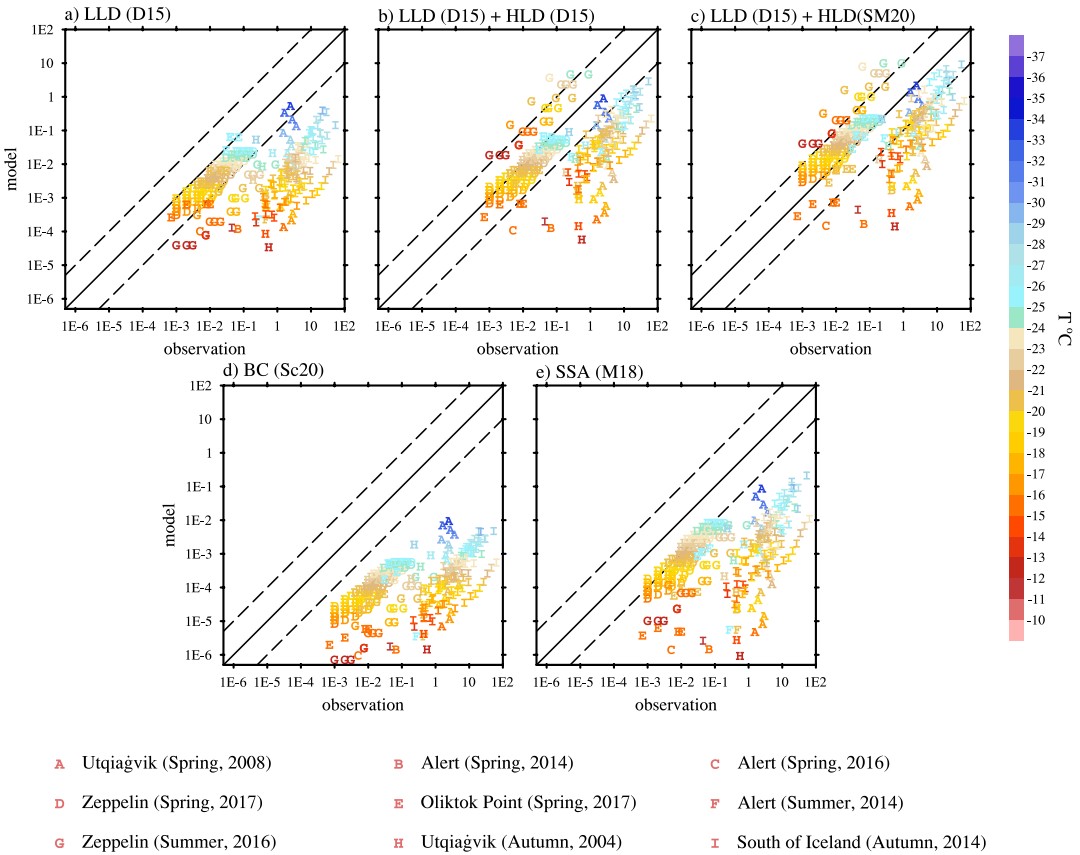

**Figure 8.** Comparison of predicted versus observed INP concentrations in the Arctic. The predicted INP concentrations are derived from a) LLD using DeMott et al. (2015; D15), b) LLD and HLD, both using D15, c) LLD using D15 and HLD using Sanchez-Marroquin et al (2020; SM20), d) BC using Schill et al. (2020; Sc20), and e) SSA using McCluskey et al. (2018; M18). SSA includes both marine organic aerosol and sea salt. Nine INP datasets are classified by symbol "A" to "I", the color of which represents the observed temperature. Details of each campaign are summarized in Table 3. Solid line in each panel represents 1:1 comparison, while dashed lines outline one order of magnitude differences. The unit for INP concentration is L$^{-1}$.

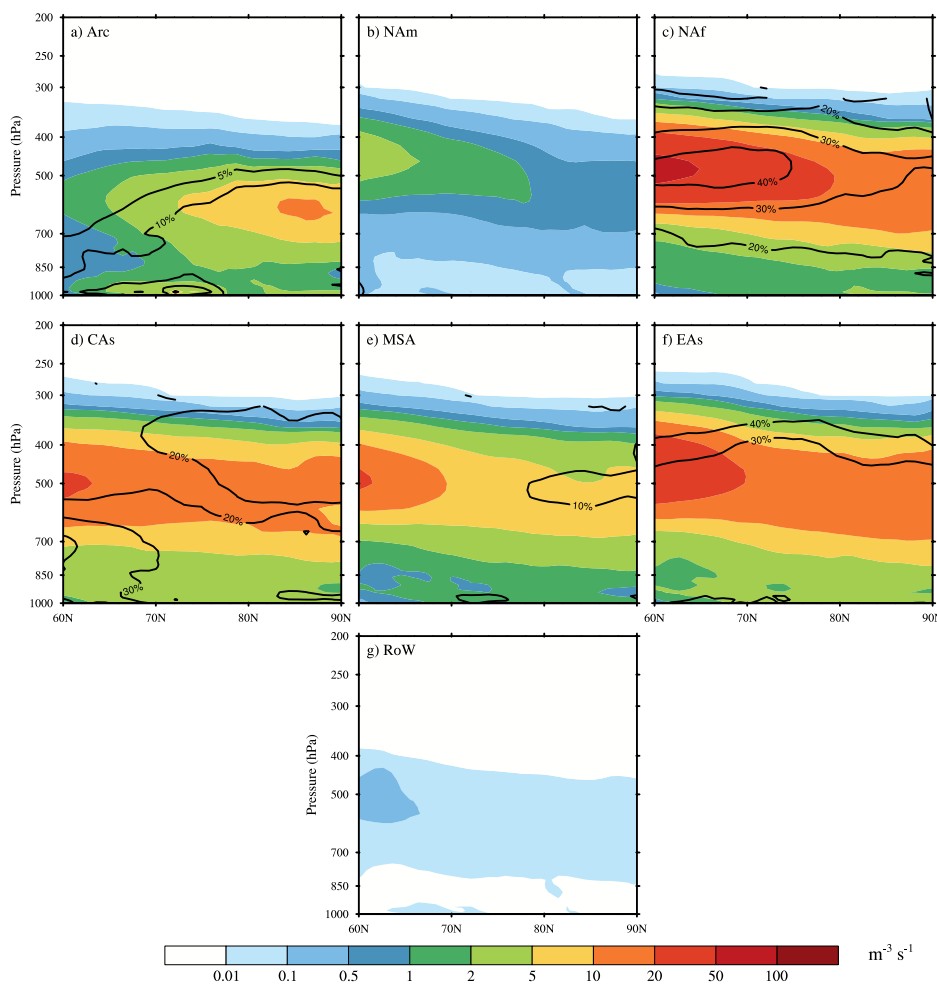

1037

**Figure 9.** Annual and zonal mean ambient mixed-phase cloud immersion freezing rates (unit: m⁻³ s⁻¹) in the Arctic (60-90°N) for the seven dust sources. Black contours are the percentage contributions from each dust source to the total immersion freezing rate.



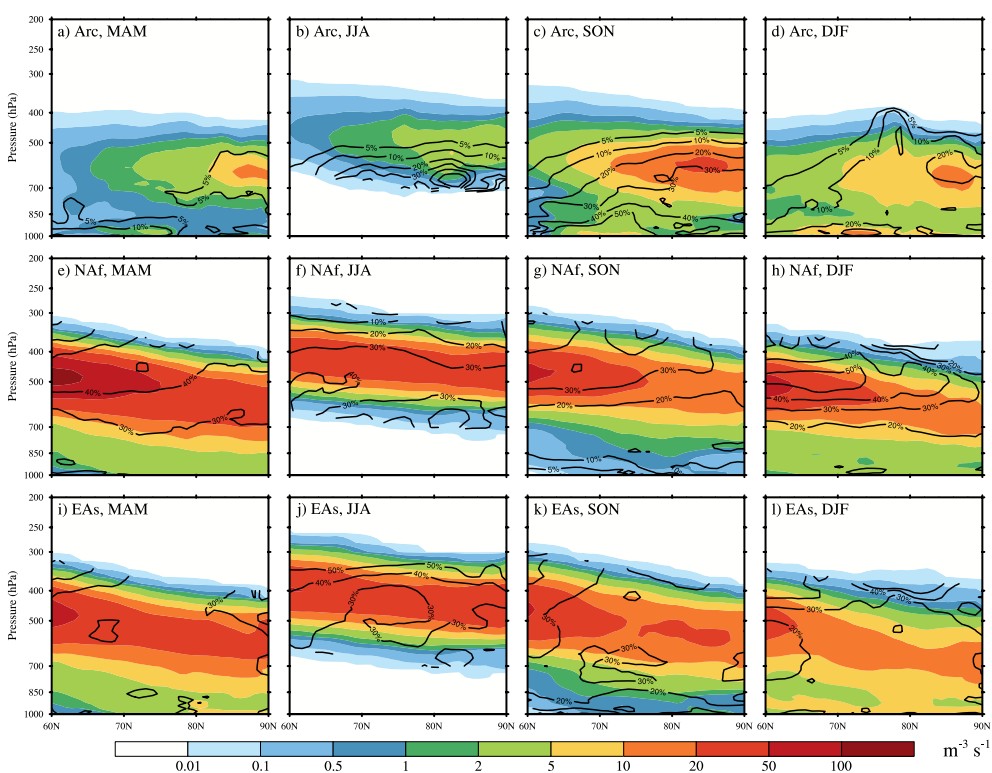

1041

**Figure 10.** Seasonal variations of the mixed-phase clouds immersion freezing rates (unit: m$^{-3}$ s$^{-1}$)

over the Arctic for dust emitted from the Arctic (top panel), North Africa (middle panel), and

East Asia (bottom panel). Black contours are the percentage contributions from each dust source

to the total immersion freezing rate in the corresponding season.

1046



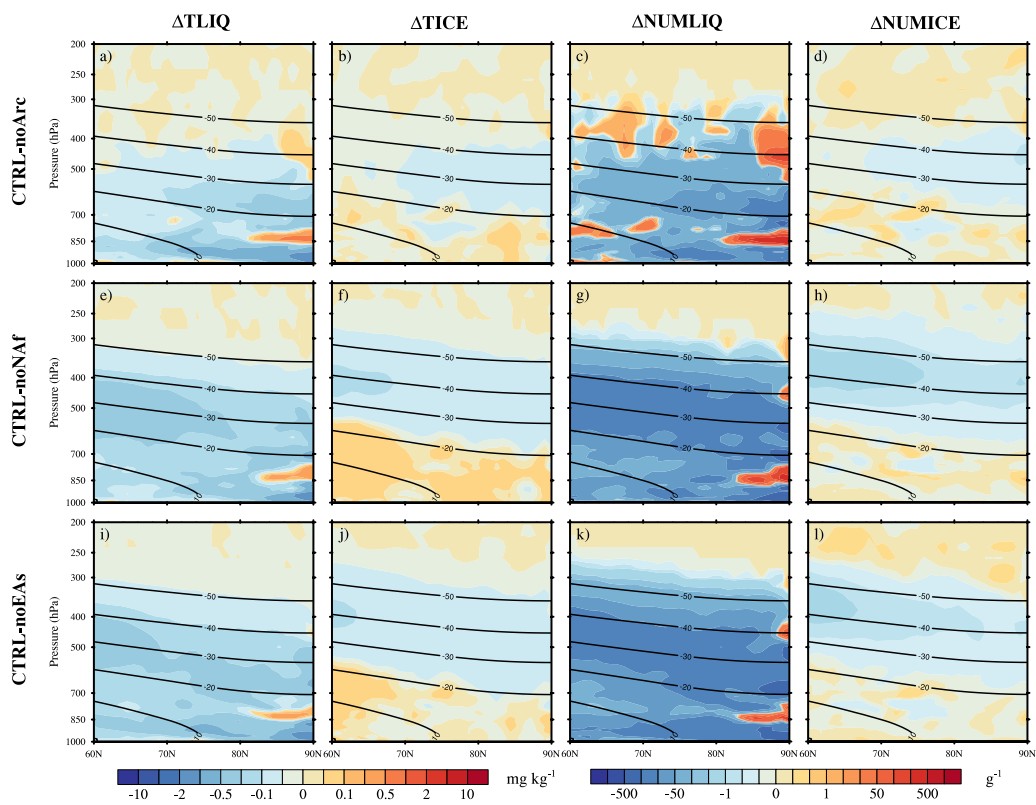

**Figure 11.** Annual and zonal mean differences in total liquid water mass mixing ratio (TLIQ), total ice mixing ratio (TICE), cloud droplet number concentration (NUMLIQ), and cloud ice number concentration (NUMICE) in the Arctic. Black contours are zonal averaged temperatures in °C. Top, middle, and bottom panels show the differences between CTRL and noArc, noNAf, and noEAs, respectively.

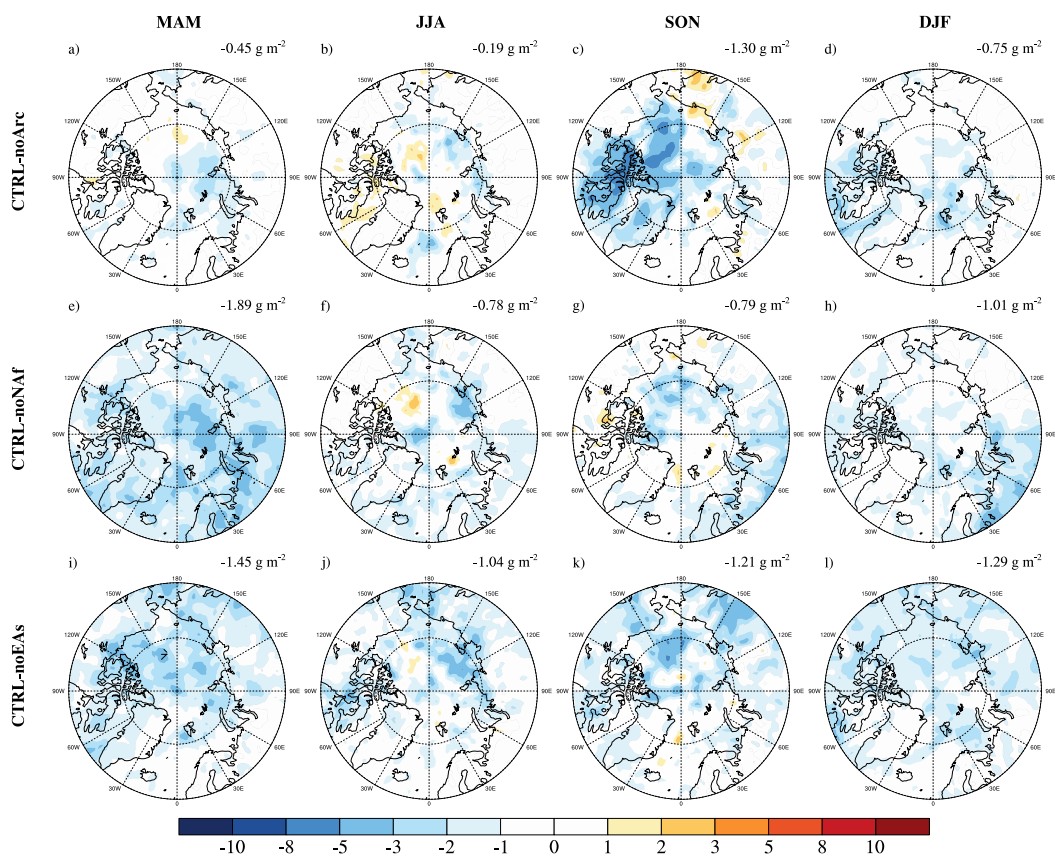

**Figure 12.** Seasonal changes in LWP (unit: g m⁻²) caused by dust INPs from the Arctic (top panel), North Africa (middle panel), and East Asia (bottom panel). The numbers are averaged LWP differences in the Arctic.

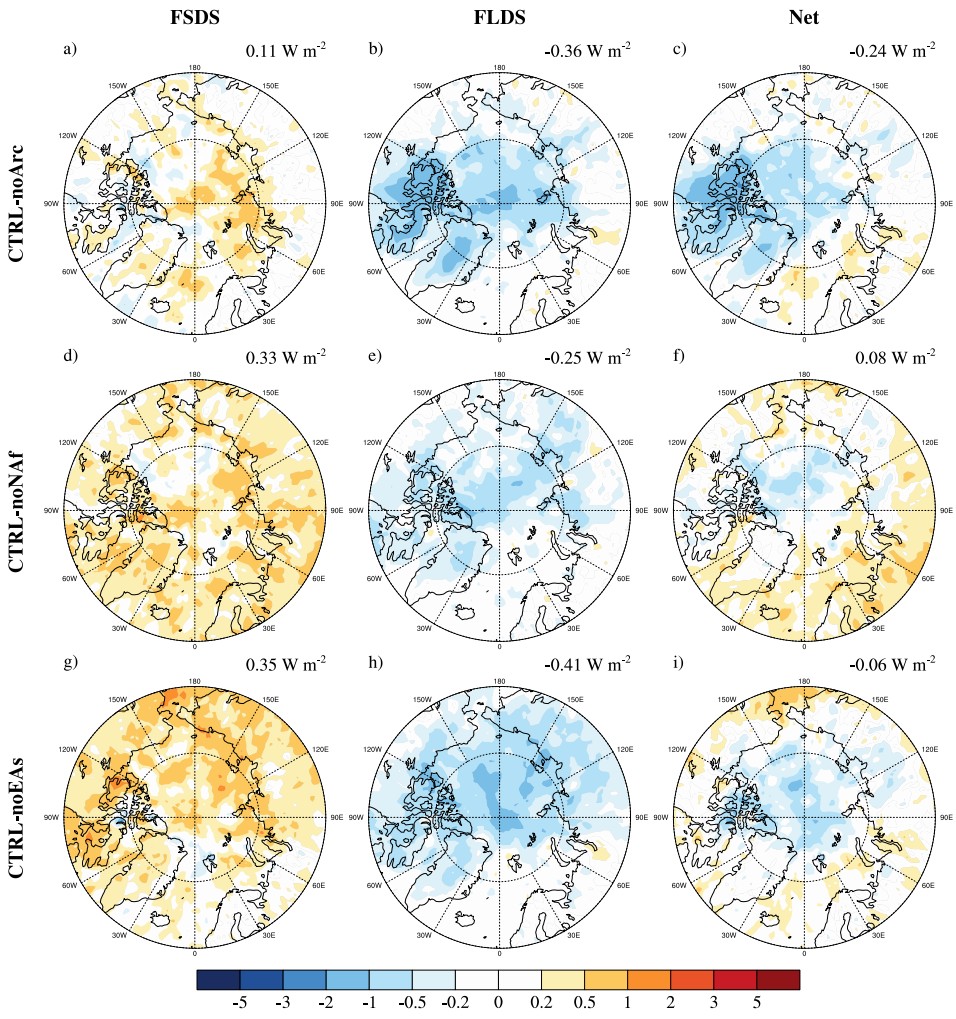

1058

**Figure 13.** Changes in annual mean downwelling radiative fluxes at the surface (unit: W m$^{-2}$)

caused by dust INPs from the Arctic (top panel), North Africa (middle panel), and East Asia

(bottom panel). Left, middle, and right panels are downwelling shortwave (FSDS), longwave

(FLDS), and net (FSDS + FLDS) radiative fluxes, respectively. The numbers are averaged

radiative flux differences in the Arctic.