# Peer review of "Transported Dust to Arctic Ice Nucleating Particles and"

_Atmospheric Chemistry and Physics, 2021_

## Author Comment (AC1)

We thank the two anonymous reviewers for their careful reading and constructive suggestions on the manuscript. Below, we explain how the comments and suggestions are addressed and make note of the revisions in the revised manuscript. The reviewers' comments are in blue color. Our replies are in black, and our corresponding revisions in the manuscript are in red.

**Responses to reviewer #1**

In this manuscript, Shi et al. use the E3SM model nudged to MERRA-2 reanalysis to determine the relative contribution of dust from six different regions to the Arctic dust load, including the local effect. They then investigate the impact of both dust from local Arctic sources (referred to as "high-latitude dust" (HLD) and dust transported from lower latitudes (referred to collectively as "low-latitude dust" (LLD) on Arctic mixed-phase clouds and the Arctic radiative budget at the surface and top of the atmosphere (TOA). The authors find that HLD, LLD from Asia and LLD from North Africa contribute to 31%, 44% and 24% of the total dust burden in the Arctic, respectively. The influence of HLD on Arctic mixed-phase clouds was found to be limited to the surface due to frequent stable thermodynamic conditions, while LLD particularly from Asia were found to influence mixed-phase clouds at colder isotherms at higher altitudes. In terms of the seasonal variations, HLD exhibited more variation and peak concentrations at summer and autumn, whereas seasonal variability was minimal for LLD, although the largest concentrations were found in spring and winter. Overall, the HLD was found to have a net cooling cloud radiative effect (CRE) at the surface to a decrease in warm liquid clouds near the surface during autumn when sunlight is relatively weak.

HLD is currently poorly characterized yet of great importance, especially in a warming world where new sources may be emitted and is thus now becoming the focus of an increasing number of studies. The work of Shi et al. is both interesting and insightful in this regard. I recommend publication of the manuscript after the authors consider some additional suggestions below.

**Reply:** We thank the reviewer for the insightful comments. We have revised the manuscript following your comments. Please see our point-to-point responses to your comments below.

1. The limited vertical transport of HLD was claimed to be due to the existence of a stably stratified Arctic lower-troposphere. I would suggest to actually quantify this using the lower tropospheric stability (e.g. as the difference in potential temperature between 850 hPa and 2m). Are there differences over sea-ice and open ocean surfaces when LTS is substantially different? Does E3SM simulate LTS in reasonable agreement with observations? Please evaluate.

**Reply:** We show the simulated LTS (defined as the potential temperature difference between 700 and 1000 hPa) from CTRL simulation and compare it to the MERRA2 reanalysis in the revised Figure S3. It shows that the LTS from CTRL agrees well with the MERRA2 reanalysis. Also, the

LTS in the Arctic is 4 to 12 K higher than the mid- and low-latitudes, which indicates a more stratified lower troposphere in the Arctic. Moreover, since the LTS over the open water is less than that over sea ice, we expect the vertical transport of HLD to be stronger over the open waters. We add discussions about these issues in the revised Section 3.2.

Line 345-354: "However, the HLD contribution decreases rapidly with height and is less than 10% above 700 hPa. This is because the lower troposphere of the Arctic is more stratified than the mid- and low latitudes, which suppresses the vertical transport of HLD. The lower tropospheric stability (LTS) from the CTRL simulation and comparison with the MERRA2 reanalysis data are shown in Figure S3. The weak HLD vertical transport in the Arctic is also reported by previous studies (Groot Zwaaftink et al., 2016, Baddock et al., 2017; Bullard, 2017). Moreover, the LTS over the Arctic sea ice is much larger than that over open ocean surface (Schweiger et al., 2008), which may lead to a stronger vertical transport of HLD over open waters. This suggests that the vertical transport of HLD may change with the sea ice reduction in a warming future."

The new Figure S3 looks as follow.

[Figure]

**Figure S3.** Annual mean (2007 to 2011) lower tropospheric stability (LTS) from MERRA2 reanalysis data and the CTRL simulation. LTS is defined as the potential temperature difference between 700 and 1000 hPa. The LTS from the CTRL simulation agrees well with the MERRA2 data.

2. Lines 38-42: The influence of these various cloud microphysical processes may also interact nonlinearly with one another and impact the phase partitioning of mixed-phase clouds as shown by Tan & Storelvmo (2016).

**Reply:** Thanks for pointing this out. We have included this statement and cited the paper in the revised manuscript:

Line 43-45: "All these processes can also interact with each other nonlinearly and impact the phase partitioning of mixed-phase clouds (Tan and Storelvmo, 2016)."

3. Source-tagging on lines 158-161: There is insufficient description of this technique in the manuscript itself. In addition to citing these references, please briefly describe the methodology and implementation of the technique. It seems that the six different regions are set up such that in addition to tagging them, they can also be separately tuned.

**Reply:** The dust source-tagging is implemented by assigning dust emitted from different sources to separate tracers. For example, in our study, there are seven tagged dust sources and thus we add seven dust tracers to the default MAM4 in E3SM. Different dust tracers are predicted independently by aerosol processes (e.g., emission, transport, and removal). Therefore, as the reviewer mentioned, each tracer can be separately tuned. We have briefly described the implementation of source-tagging in the revised manuscript:

Line 167-170: "In this method, dust emission fluxes from different sources are assigned to separate tracers and transport independently, so that dust originating from different sources can be tracked and tuned separately in a single model experiment."

4. Is aging of aerosols and the addition of coatings of pollutants that may modify the ice-nucleation efficiency of dust INPs represented in E3SM?

**Reply:** Earlier studies have found that chemical aging or coating in the atmosphere may change the ice nucleating ability of dust (Boose et al., 2016; Kulkarni et al., 2014). Our model does not consider this effect explicitly. In the CNT parameterization (please note we add the CNT scheme to the INP comparison in Section 3.3 following reviewer #2's comment), the depression of immersion freezing point by sulfate coating is considered. However, this depression effect has no differences for HLD and LLD, because dust aerosols are assumed to be internally mixed with sulfate within an aerosol mode in the MAM4 aerosol module (Liu et al., 2016) (see also in Text S2.1). The other dust ice nucleation parameterizations we used (i.e., SM20 and D15) may have already taken the aging/coating effect into account implicitly. For HLD, SM20 was derived from freshly emitted aerosol samples collected close to Iceland. For LLD, D15 included Saharan and Asian dust data collected over the Pacific Ocean basin and US Virgin Islands, respectively, which are far away from the corresponding LLD sources (we add more details about D15 and SM20 in Text S2). We have clarified this and added the discussion in the revised manuscript:

Line 436-442: "In addition, we do not explicitly represent the potential ice nucleation ability differences in freshly emitted HLD and long-range transported LLD caused by the aging and the coatings of pollutants (Kulkarni et al., 2014; Boose et al., 2016). However, D15 and SM20 may already take the aging effect into account implicitly. Because D15 is based on the Saharan and Asian dust data collected over the Pacific Ocean basin and US Virgin Islands, respectively, which are far away from the corresponding LLD sources, while SM20 is derived from the freshly emitted Icelandic HLD, which is subjected to less aging effect."

5. Given the large discrepancy between model and observations in Alert, the authors should consider utilizing long-term observations of dust available in Alert as described in Sirois and Barrie (1999).

**Reply:** Thanks for the suggestion. We replaced the Alert dust concentration measurements with Sirois and Barrie (1999) and updated Figures 2, 3, and previous Figure S8 (now Fig. S11). The dust concentrations from this long-term observations (1980 to 1995) are 5 to 10 times higher than those from Fan (2013), which leads to a better agreement in the comparison of annual mean dust concentrations at Alert (Figure 2b in the revised manuscript). However, the simulated results still show large high bias in the summertime (Figure 3b and S11 in the revised manuscript). So, our main conclusion related to the Alert dust comparison does not change. We still attribute the large discrepancies to the limitation of the dust emission parameterization.

The revised Figure 2 looks as follow (only the red dot representing annual mean dust surface concentration at Alert on Figure 2b is changed).

[Figure]

The revised Figure 3 looks as follow (only Figure 3b is changed).

[Figure]

The previous Figure S8 (now Figure S11) looks as follow.

[Figure]

6. Are the observations and simulated AOD, dust concentrations and deposition flux directly comparable? For example, observations of cloud properties cannot be directly compared with remote sensing observations without a simulator to account for differences in the definitions of these quantities. One would expect the same for aerosol properties as well.

**Reply:** This is a valid point. Comparing the observations directly with simulated dust properties leads to some uncertainties. For AOD, the AERONET measurements are biased towards clear-sky conditions due to the cloud-screening procedure (Smirnov et al., 2000), while the simulated results are under all-sky conditions. For dust concentrations and dust deposition fluxes, the cut-off size of the measurements may be a source of uncertainties in the comparisons. The simulated dust particles are mainly smaller than 10 μm, while the cut-off sizes of the measurements vary from several micrometers to several tens of micrometers. We tried to address this bias for the dust concentration comparison at Trapper Creek (Figure 3c; cut-off size at 2.5 μm) and during ARCTAS flight campaign (Figure 4; cut-off size at 4 μm) by applying the same cut-off sizes to the simulated dust concentrations. However, it is hard for us to do similar things for other comparisons, because the cut-off sizes of these observations are either unclear (e.g., the dust deposition fluxes dataset) or much larger than the simulated dust size range (e.g., many dust concentration measurements have a cut-off size of 40 μm). However, the cut-off size may not be a significant source of bias in the latter situation, because most of the dust concentration data corresponds to measurements at remote stations where most of the super coarse dust particles (>10 μm) cannot reach due to the quick sedimentation. Finally, in addition to the uncertainties mentioned above, the comparisons have representative issues caused by comparing an observational station with a global model grid that has a size of ~100 km. Some of the comparisons also have systematic errors because the measurements were for a different time period than that of the model simulation. Nevertheless, despite all the uncertainties, the direct comparisons have been widely used by previous studies of dust properties (e.g. Huneeus et al., 2011; Kok et al., 2014; Albani et al., 2014).

We summarize the major biases of the comparisons in the revised manuscript:

Line 203-205: "We note that the AERONET AOD measurements are biased towards clear-sky conditions due to the cloud-screening procedure (Smirnov et al., 2000)."

Line 216-220: "We note that the comparisons are subject to representative biases caused by comparing an observational station with a global model grid point (with a horizontal resolution of ~ 100 km). The comparisons of dust concentration and deposition flux also have systematic errors because the measurements were for a different time period than that of the model simulation."

7. The discrepancy (up to almost twice) between this study and previous studies in terms of the contribution of North African dust to the Arctic dust burden is quite large. Potential reasons are listed on lines 308-310: Of these processes, which process dominates?

**Reply:** The wet removal may be one of the dominant processes. This process depends on the model representation of clouds and precipitation, which have large discrepancies among different models. The dust emission parameterization may be another key factor contributing to the discrepancies. For example, the fraction of total dust emission flux from each source varies if using different dust emission parameterizations. The spatial distribution of dust emission "hot spots" may also be different, which is likely to influence the transport efficiency of dust emitted from each source. Another factor that may have some impacts is the size distribution of the dust emission. In our model, we have more dust emitted in the coarse mode (according to Kok (2011)) than earlier studies, which leads to a shorter dust lifetime in our simulations. More detailed comparisons between our work and previous studies are needed for a definite answer, which is beyond the scope of this study.

8. Figures 12 and 13: It would be useful to compare how E3SM simulates Arctic CREs at the surface and TOA (e.g. comparing with the NASA CERES instrument), and how HLD vs LLD contributes to biases in the Arctic CREs in an additional column. Similarly for the LWP. The MODIS simulator can be used for the sunlit months.

**Reply:** Thanks for this good suggestion. We evaluate the simulated Arctic LWP and radiative fluxes with MODIS and CERES, respectively (new Figure 14 in the revised manuscript). Two MODIS datasets are used, including the standard product (Platnick et al., 2003; P03) and an improved one (Khanal et al., 2020; K20) that corrected the positive bias in the Arctic in P03. We rerun all the four simulations to turn on the MODIS simulator for the LWP comparison. Please note the reruns are only conducted for two years (2007 and 2008), due to the limit in the computer resources. The original five-year simulations are used for the CERES comparison.

According to Fig. 14a, the simulated Arctic LWP from all the simulations are lower than P03 but higher than K20. The differences among simulations are very small compared to their discrepancies with MODIS observations. But according to the numbers shown above the bar charts, including dust INPs from each of the three sources decreases the LWP (i.e., CTRL has less LWP than the other simulations), which makes comparisons slightly better as compared to K20.

The comparisons of downwelling radiative fluxes and TOA cloud radiative forcing are shown on Figs. 14b-e. Compared to CERES, all the simulations underestimate FSDS with too strong SWCF and overestimate FLDS with too strong LWCF, which likely points to the biases of modeled clouds (e.g., too much LWP as compared to K20). Similar to the LWP comparison, the differences among simulations are very small compared to the discrepancies with CERES observations. However, we do see some improvements after including dust INPs from each of the three sources (i.e., the results from CTRL are closer to the CERES results than the other three simulations).

Overall, including HLD or LLD INPs do not contribute a lot to reduction of the biases in simulating the LWP and CREs in the AMPCs. However, the representation of AMPCs in global climate

models is associated with multiple cloud macro- and microphysical processes, and large-scale dynamics (Morrison et al., 2012). As mentioned by the reviewer in comment 2, these processes interact with each other non-linearly. Therefore, even though including HLD or LLD INPs do not improve the representation of AMPCs significantly in our model, a good representation of dust INPs, especially including HLD INPs, could still be of great importance for parameterizing AMPCs.

The new Figure 14 looks:

[Figure]

**Figure 14.** a) Annual mean Arctic averaged LWP over ocean for the MODIS observations (2007-2009) and the four simulations (2007-2008). Two MODIS datasets are used, including the standard product (Platnick et al., 2003; P03) and an improved one (Khanal et al., 2020; K20). The MODIS simulator is used to calculate the simulated LWP. b) - e) Annual mean Arctic averaged b) FSDS,

c) FLDS, d) SWCF, and e) LWCF for the CERES observation (2007-2011) and the four simulations (2007-2011).

We added discussions in Section 3.4 to address the reviewer's comment:

Line 534-558: "Finally, we evaluate the model performance in simulating the Arctic LWP and radiative fluxes against the Moderate Resolution Imaging Spectroradiometer (MODIS) LWP (Platnick et al., 2003) and the Cloud and the Earth's Radiant Energy System Energy Balanced and Filled Edition 4.1 (CERES-EBAF Ed4.1) products (Loeb et al., 2018; Kato et al., 2018), respectively (Figure 14). Two MODIS datasets are used, including the standard product (Platnick et al., 2003; P03) and an improved one (Khanal et al., 2020; K20) that corrected the positive bias in the Arctic in P03. The MODIS simulator is used for the LWP comparison. According to Fig. 14, the simulated LWP from the four experiments are lower than P03 but higher than K20. All the four experiments also underestimate FSDS with too strong SWCF and overestimate FLDS with too strong LWCF, which likely points to the biases of modeled clouds (e.g., too much LWP as compared to K20). The differences among the model experiments are very small compared to their discrepancies with observations. We notice including dust INPs from the three sources decreases the simulated LWP (i.e., CTRL has less LWP than the other experiments) (Figure 14a), which makes the model performance better if compared to K20. Moreover, it shows noticeable improvements in simulating both surface and TOA radiative fluxes after including dust INPs from each of the three sources (i.e., the results from CTRL are closer to the CERES results than the other three experiments) (Figure 14b-e).

Overall, including HLD or LLD INPs do not contribute a lot to the reduction of biases in simulating the LWP and radiative fluxes in the AMPCs. However, the representation of AMPCs in global climate models is associated with multiple cloud macro- and microphysical processes, and large-scale dynamics (Morrison et al., 2012) (see more discussion in Section 4), which interact with one another non-linearly. Therefore, even though including HLD or LLD INPs do not improve the representation of AMPCs significantly in our model, a good representation of dust INPs, especially including HLD INPs, could still be of great importance for parameterizing AMPCs in the model."

9. Comparison with CALIOP: Why use observations from 2007-2009? The record extends well beyond that and the 2007 observations are partially impacted by the change in the tilt of the nadir-viewing angle. Also, what CALIOP product was used and what was the version of the product? Arctic aerosol layers are frequently too tenuous to be detected by CALIOP and are also furthermore impacted by the presence of clouds that can interfere with the cloud-aerosol discrimination algorithm.

**Reply:** We use the CALIPSO dust extinction dataset developed by Luo et al. (2015a, 2015b). Luo et al. (2015a) developed a new method for dust separation from other aerosol types to derive the dust backscatter coefficient in the lidar equation inversion stage using CAL-L1B data, which has less uncertainties than doing the separation based on lidar inversion products (i.e., CAL-L2) in

previous studies (e.g., Amiridis et al., 2013; Yu et al., 2015). Luo et al. (2015b) further developed a new dust identification method by using combined lidar-radar cloud masks from CloudSat and CALIPSO, which significantly improves the detection of optically thin dust layer, especially in the Arctic. We use both the new dust separation method (Luo et al., 2015a) and the new dust identification method (Luo et al., 2015b) to produce the nighttime dust extinction dataset.

We use the retrievals during 2007-2009 because of the data availability (lidar-radar cloud masks). Due to the battery anomaly on April 17[th], 2011 the CloudSat stopped collecting data for ~1 year and since then continued to only operate during the sunlit portion of the orbit with degraded overlap with CALIPSO. Therefore, lidar-only cloud masks are needed for retrieving nighttime dust extinction, which have been in the development.

We notice that the nadir angle was increased from 0.3º to 3º to reduce specular returns from clouds containing horizontally oriented ice crystals in November 2007. According to the official document (https://asdc.larc.nasa.gov/documents/calipso/TiltModeGeometry.pdf), the optical properties reported for the measurements of aerosols and water clouds are not expected to change as a result of the change in pointing angle. However, the properties reported for individual ice clouds will change by varying amounts, which may contribute to retrieval uncertainties.

We added some descriptions about the CALIPSO data used in this study in Section 3.1:

Line 266-268: "The Luo et al. (2015a, b) data set has improvements in dust separation from other aerosol types and thin dust layer detection in the Arctic than the standard CALIPSO product (Winker et al., 2013)."

Typographical error:

- Line 137: "hour" should be "hours"

**Reply:** It is corrected. Thanks.

---

## Author Comment (AC2)

We thank the two anonymous reviewers for their careful reading and constructive suggestions on the manuscript. Below, we explain how the comments and suggestions are addressed and make note of the revisions in the revised manuscript. The reviewers' comments are in blue color. Our replies are in black, and our corresponding revisions in the manuscript are in red.

**Responses to reviewer #2**

Shi et al. present a global modeling study to estimate the contribution of dust emitted at high latitudes as source of ice nucleating particles in the Arctic. They added a source tagging technique for dust from different regions to accomplish this. The role of dust in the climate system is an important topic, especially in high latitudes, which are experiencing a rapid change in climate. The paper makes an important contribution to this topic, since high-latitude dust contributions are largely unknown. At the same time the paper highlights several challenges -- simulating the global distribution of dust itself, but also estimating the concentration of ice nucleating particles based on the simulated dust distribution. The paper fits well within the scope of ACP, and I recommend the paper to be published after the following comments are addressed:

**Reply:** We thank the reviewer for the constructive comments, which have improved the quality of our manuscript. We have revised the manuscript following your comments.

1. Line 124: More detail is needed to describe the ice-nucleation parameterization that is used in the simulations (I believe this is the soccer-ball model?). How is dust represented using this parameterization? Are the same model parameters applied not matter from which region the dust comes from (i.e. different mineralogical composition is ignored)?

**Reply:** The ice nucleation parameterization used in the simulations is the classical nucleation theory (CNT) parameterization. Wang et al. (2014) implemented the CNT parameterization based on Hoose et al. (2010) into the Community Atmosphere Model (CAM) but improved it by using a probability density function (PDF) of contact angles for immersion freezing of dust instead of a single contact angle. In addition, CNT treats the deposition and contact ice nucleation on dust particles. Please note this scheme is not the soccer-ball model implemented by Wang and Liu (2014). Dust particles originated from different sources are treated the same in the model (i.e., the same PDF of contact angles used). We do not consider the differences in mineralogical composition. We provide more details about the CNT parameterization in Text S2.1. Please see more details in our responses to your comment #4.

2. Related to this, in section 3.3, different ice-nucleation parameterizations are used for the comparison with measurements. It seems that the default parameterization should be part of this comparison. I suggest adding this to this section.

**Reply:** Thanks for the suggestion. We agree with you on this. The default CNT parameterization is now included in the INP comparison in Section 3.3. We also revised Figure 8 accordingly. Overall, the results from the CNT comparison (new Figures 8a-8c) are consistent with those from the comparison of other dust ice nucleation parameterizations (new Figures 8d-8f) – including the contribution from HLD improves the model performance in simulating Arctic INP concentrations. The CNT parameterization produces 5 to 10 times higher INP concentrations than the other two schemes (D15 and SM20) at moderately cold temperatures (-22 to -28°C), while it has a significant underprediction of INP concentration at warm temperatures (T > -18°C).

The revised Figure 8 looks as follow.

[Figure]

| | | |
|---|---|---|
| **A** Utqiaġvik (Spring, 2008) | **B** Alert (Spring, 2014) | **C** Alert (Spring, 2016) |
| **D** Zeppelin (Spring, 2017) | **E** Oliktok Point (Spring, 2017) | **F** Alert (Summer, 2014) |
| **G** Zeppelin (Summer, 2016) | **H** Utqiaġvik (Autumn, 2004) | **I** South of Iceland (Autumn, 2014) |

**Figure 8.** Comparison of predicted versus observed INP concentrations in the Arctic. The predicted INP concentrations are derived from a) LLD using classical nucleation theory (CNT), b) LLD and HLD, both using CNT, c) LLD using CNT and HLD using Sanchez-Marroquin et al (2020; SM20), c) LLD using DeMott et al. (2015; D15), d) LLD and HLD, both using D15, e) LLD using D15 and HLD using SM20, f) BC using Schill et al. (2020; Sc20), and g) SSA using McCluskey et al. (2018; M18). SSA includes both marine organic aerosol and sea salt. Nine INP datasets are classified by symbol "A" to "I", the color of which represents the temperature reported in the observations. The observations for datasets "A", "C", "E", "H" are monthly mean values. Samples for datasets "D", "G", "I" are selected randomly and only 15% of them are plotted. Details of each campaign are summarized in Table 3. The modelled INP concentrations are diagnosed using the observed temperatures and monthly averaged aerosol properties of the corresponding month from year 2007 to 2011. The INP concentrations for CNT are defined as the CNT immersion freezing rate integrated by 10 s, following Hoose et al. (2010) and Wang et al. (2014). Solid line in each panel represents 1:1 comparison, while dashed lines outline one order of magnitude differences. The unit for INP concentration is $L^{-1}$.

We added a brief discussion regarding CNT in Section 3.3:

Line 395-401: "Overall, only including LLD as INPs results in up to four orders of magnitude underprediction compared to observations (Figure 8a and 8d), while taking into account the contribution from HLD greatly improves the model performance by increasing the simulated dust INP concentrations (Figures 8b, 8c, 8e, and 8f). The CNT parameterization produces 5 to 10 times more INP concentrations than the other two schemes at moderately cold temperatures (-22 to -28℃), while it has a significant underestimation of observed INP concentrations at warm temperatures (T > -18℃) (also see Figure S4)."

We also added a new figure (new Figure S4) in the supporting information showing the relationship of simulated INP concentrations and the temperatures. The new Figure S4 looks as follow.

[Figure]

**Figure S4.** Simulated INP concentrations as a function of temperature. The simulated INP concentrations are derived from a) LLD using classical nucleation theory (CNT), b) LLD and HLD, both using CNT, c) LLD using CNT and HLD using Sanchez-Marroquin et al (2020; SM20), c) LLD using DeMott et al. (2015; D15), d) LLD and HLD, both using D15, e) LLD using D15 and HLD using SM20, f) BC using Schill et al. (2020; Sc20), and g) SSA using McCluskey et al. (2018; M18). The temperature of each data point is also shown by its color. Nine INP datasets are classified by symbol "A" to "I". This figure is based on the same simulated INP concentrations that used from Figure 8.

3. Line 134: What was the rational for choosing the time period 2006 to 2011? (and not for example a more recent time period for example)

**Reply:** The primary reason for choosing year 2007 to 2009 is to match the time period for the CALIPSO dust retrievals (Luo et al., 2015a, b). We regard the CALIPSO dust extinction as a very valuable dataset because it gives vertical profiles for almost the entire Arctic region. We then extended the simulations to year 2011 to reduce noises that shows up in the dust INP effects. Also, we have tuned the global dust optical depth to 0.030±0.005, which is based on the estimates by Ridley et al. (2016) over year 2004-2008. Therefore, considering the possible decadal changes in global dust distribution, a second reason for choosing this time period (not a more recent one) is that we want the simulations to be conducted in a time period that is closer to that from Ridley et al. (2016).

We note that choosing this time period leads to biases in comparing our model results with measurements conducted in other time periods. This may be particularly true for the INP comparisons, because 7 out of 9 INP datasets we use were conducted after year 2012 and the year-to-year variability may be large for the measured INPs in the Arctic. However, considering the large uncertainties with the INPs, this year-to-year variability is of secondary order.

4. Line 150: Similar to comment 1, I recommend explaining more detail (i.e. equations) about the dust emission parameterization as well as the source tagging procedure. INP parameterization, dust emission parameterization and source tagging are central to this paper, so even though they are described in other papers, it will be helpful for the reader to have the information easily available. This could go in an Appendix or even the SI.

**Reply:** Thanks for the suggestion. The source tagging procedure is implemented by assigning dust emitted from different sources to separate tracers. It does not involve changes in equations. We add a few sentences to explain the implementation of the source-tagging technique in the revised manuscript:

Line 167-170: "In this method, dust emission fluxes from different sources are assigned to separate tracers and transport independently, so that dust originating from different sources can be tracked and tuned separately in a single model experiment."

We provide equations for the dust emission parameterization and ice nucleation parameterizations in the revised SI (Text S1 and S2):

**"Text S1: K14 dust emission parameterization**

Kok et al. (2014a, b) (K14) is a physically based dust emission scheme that removes the need to use an empirical dust soil erodibility map in other parameterizations (e.g., Zender et al., 2003). The vertical dust emission flux, $F_d$ (kg m$^{-2}$ s$^{-1}$) is given by

$$F_d = C_d f_{bare} f_{clay} \frac{\rho_a(u_*^2 - u_{*t}^2)}{u_{*st}} \left(\frac{u_*}{u_{*t}}\right) C_\alpha^{\frac{u_{*st} - u_{*st0}}{u_{*st0}}}, \ (u_* > u_{*t}), \tag{S1}$$

where $C_d$ is the dimensionless dust emission coefficient, $f_{bare}$ is the fraction of the surface consisting of bare soil, $f_{clay}$ is the soil clay fraction, $\rho_a$ (kg m$^{-3}$) is the air density, $u_*$ (m s$^{-1}$) is the soil friction velocity, $u_{*t}$ (m s$^{-1}$) is the threshold of soil friction velocity above which saltation occurs, $u_{*st}$ (m s$^{-1}$) is the soil threshold friction velocity standardized to standard atmospheric density, $u_{*st0}$ (m s$^{-1}$) is the standardized threshold friction velocity of an optimally erodible soil, and $C_\alpha$ is the dimensionless constant scaling the fragmentation exponent ($\alpha$).

**Text S2: Ice nucleation parameterizations**

In this section, we introduced five ice nucleation parameterizations used in this study. They can be classified into two types: the stochastic approach, which treats ice nucleation as a time-dependent process, and the deterministic approach, which assumes that ice nucleation is time-invariant and only depends on temperature and aerosol properties. In this study, the CNT parameterization follows the stochastic approach, while the other four parameterizations follow the deterministic approach.

**S2.1 CNT parameterization**

The classical nucleation theory (CNT) scheme is used for heterogeneous ice nucleation in mixed-phase clouds in EAMv1 simulations. This parameterization was first implemented in a global climate model by Hoose et al. (2010), and further improved by Wang et al. (2014) by introducing a probability density function of contact angles ($\alpha$-PDF) for immersion freezing of natural dust. In CNT, immersion/condensation, contact, and deposition nucleation on dust and BC are treated. The rate of heterogeneous nucleation per aerosol particle and time, $J_{het}$, is expressed by

$$J_{het} = \frac{A' r_N^2}{\sqrt{f}} \exp\left(\frac{-\Delta g^{\#} - f \Delta g_g^o}{kT}\right), \tag{S2}$$

where $A'$ is a prefactor, $r_N$ is the aerosol particle radius, $f$ is a form factor describing the aerosol's ice nucleating ability, $\Delta g^{\#}$ is the activation energy, $\Delta g_g^o$ is the homogeneous energy of germ formation, $k$ is the Boltzmann constant, and T is the temperature in K. The factor, $f$, is a function of contact angle, $\alpha$, in the form,

$$f = \frac{1}{4}(2+m)(1-m)^2, \tag{S3}$$

where $m \equiv cos\alpha$. The contact angle is assumed to follow a log-normal distribution in the form,

$$p(\alpha) = \frac{1}{\alpha\sigma\sqrt{2\pi}} \exp\left(-\frac{(\ln(\alpha) - \ln(\mu))^2}{2\sigma^2}\right), \tag{S4}$$

where $\mu$ is the mean contact angle and $\sigma$ is the standard deviation.

We do not consider the differences in the mineralogical composition in different dust sources. Thus, dust particles originated from different sources are treated the same in the CNT (i.e., same contact angle distribution). The parameterization considers the immersion freezing point depression by coating of sulfate aerosols (Hoose et al., 2010). However, this effect has no

differences for HLD and LLD, because aerosol species (e.g, dust, sulfate) are assumed to be internally mixed within an aerosol mode in the MAM4 aerosol module (Liu et al., 2016).

**S2.2 D15 parameterization**

The DeMott et al. (2015; D15) parameterization is a dust immersion freezing ice nucleation parameterization derived from a combination of laboratory and field data. The laboratory data are from ice nucleation experiments on Saharan and Asian desert dust using the Aerosol Interaction and Dynamics in the Atmosphere chamber. The field data were collected over the Pacific Ocean basin and US Virgin Islands, which are dominated by Asian and Saharan desert dust, respectively. Thereby, D15 can be regarded as a LLD ice nucleation parameterization in our study, though it is also applied to HLD in Figure 8f for sensitivity studies. In D15, dust INP number concentration, $n_{INP}$ (std L$^{-1}$), is related to temperature, $T_k$ (K), and the number concentration of dust particles larger than 0.5 $\mu m$, $n_{a>0.5\mu m}$ (std cm$^{-3}$), in the form,

$$n_{INP}(T_k) = (cf)(n_{a>0.5\mu m})^{\alpha(273.16-T_k)+\beta}\exp(\gamma(273.16 - T_k) + \delta), \tag{S5}$$

where $cf = 3$, $\alpha = 0$, $\beta = 1.25$, $\gamma = 0.46$, and $\delta$ = -11.6.

**S2.3 SM20 parameterization**

The Sanchez-Marroquin et al. (2020; SM20) parameterization is based on aircraft-collected freshly emitted Icelandic dust samples and thus is treated as a parameterization for HLD in our study. It is an immersion freezing ice nucleation scheme formulated in terms of the ice-nucleating active surface site density (INAS). The total INP concentration, $n_{INP}$ (L$^{-1}$), is given by

$$n_{INP} = n_{HLD}\{1 - \exp[-S_{ae}n_s]\}, \tag{S6}$$

where $n_{HLD}$ (L$^{-1}$) is the number concentration of HLD, $S_{ae}$ (m$^2$) is the surface area of a single HLD particle, and $n_s$ (m$^{-2}$) is the density of active sites. $n_s$ is in the form

$$n_s(T) = 10^{-0.0337-0.199T}, \tag{S7}$$

where T is temperature in °C.

**S2.4 Sc20 parameterization**

The Schill et al. (2020; Sc20) parameterization is an INAS-based immersion freezing ice nucleation parameterization based on smoke from western US wildfires and grassland prescribed burns. It is an ice nucleation parameterization for biomass burning black carbon (BC), but we apply it to BC from both biomass burning and fossil fuel combustion. The total INP concentration is given by the same equation as Eq.(S6), except $n_{HLD}$ is replaced by $n_{BC}$, which is the number concentration of BC. The $n_s$ fit for Sc20 is given by

$$n_s(T) = \exp(1.844 - 0.684T - 0.00597T^2), \tag{S8}$$

where T is temperature in °C.

**S2.5 M18 parameterization**

The McCluskey et al. (2018; M18) parameterization is an INAS based immersion freezing ice nucleation parameterization for sea spray aerosols (SSAs; includes sea salt and marine organic aerosol) derived from pristine marine air mass measurements at the Mace Head Research Station.

The total INP concentration is given by the same equation as Eq.(S6), except $n_{HLD}$ is replaced by $n_{SSA}$, which is the number concentration of SSA. The $n_s$ fit for M18 is given by

$$n_s(T_k) = \exp\left(-0.545(T_k - 273.15) + 1.0125\right), \qquad\qquad \text{(S9)}$$

where $T_k$ is temperature in K."

    5. Line 155-157: Are the differences between the mass assignments to modes in Z03 and Kok (2011) significant?

**Reply:** Global models usually have large uncertainties in simulating dust. To add some constraints to the global dust fields, the global averaged dust optical depth (DOD) is usually tuned to be within the range of the observational estimate (0.030±0.005) by Ridley et al. (2016). This is what we did for our CTRL experiment and thus the global averaged DOD does not change much by changing the mass assignments.

When we change the mass assignments from Z03 to Kok (2011) by shifting dust mass from the accumulation mode to the coarse mode, we need to raise the global dust emission fluxes to match the global averaged DOD because the coarse mode dust contributes less to total DOD than the accumulation mode dust per mass base. Therefore, dust emission fluxes and dust burden both increase after changing the emission size distribution from Z03 to K11. We note the dust burden changes are more obvious near the dust sources, because the coarse dust particles are removed readily during transport. The remote regions (the Arctic) are less affected by changing the dust emission size distributions.

    6. Explanation of quantities shown in the figures: Please add information in the caption over what time period the model results were averaged. For example, in Fig 3, is this the average over the entire simulated period (2006-2011)? If so, what is the year-to-year variability? And is it the same time period for the observations? What does the grey band represent?

**Reply:** In Figure 3, the model results are averaged from 2007 to 2011. It is not the same time period for the observations. We add the year-to-year variability of the modelled total dust concentration (the pink shade) to Figure 3. The grey band represents the standard deviation of observations. We revise Fig. 3 and its caption following your suggestions.

[Figure]

**Figure 3.** Comparison of measured (black solid line, with gray shade representing standard deviation) and simulated (pink solid line, with pink shade representing year-to-year variability) monthly mean dust surface concentration at three high latitude stations – a) Heimaey, b) Alert, and c) Trapper Creek. The model results are averaged from year 2007 to 2011. Contributions from seven tagged sources are shown by colored dashed lines. The locations of the three stations are shown in Figure 2d. The measurements at Heimaey (Prospero et al., 2012), Alert (Sirois and Barrie, 1999), and Trapper Creek (IMPROVE) are averaged for the years 1997 to 2002, 1980 to 1995, and 2007 to 2011, respectively. The dust concentrations at Trapper Creek only include particles with diameter less than 2.5 μm. The other two stations include dust over the whole size range.

7. Fig 4: The caption mentions that the model results were averaged w.r.t time (2007-2011) and space. I suggest showing some measure of variability, with respect to time and space, to be better able to judge the agreement with the observations.

**Reply:** Following your suggestion, we have added the standard deviation with respect to time and space for the simulated total dust concentrations on Fig. 4. The revised figure and figure caption looks as follow:

[Figure]

**Figure 4.** Comparison of vertical dust concentrations from ARCTAS flight observations (Jacob et al., 2010) (black circle) and CTRL simulation (pink solid line) in a) April and b) July. We show median values for observations at each level. The maximum and minimum of the measurements at each level are shown by black lines. Contributions from the seven tagged sources in CTRL are shown by colored dashed lines. The ARCTAS dust mass concentrations are derived from measured calcium and sodium concentrations. The measurements data are processed using the same method as Breider et al. (2020). Briefly, we assume a calcium to dust mass ratio of 6.8% and further correct the calcium concentrations for sea salt by assuming a calcium to sodium ratio of 4%. Only measurements obtained north of 60°N are used for the analyses. The low-altitude observations near Fairbanks, Barrow, and Prudhoe Bay are removed. Also, data from below 1 km on 1, 4, 5, 9 July is removed to exclude the influence of wildfire. The ARCTAS flight campaign was conducted in 2008, while the modeled vertical profiles are averaged for each April and July from 2007 to 2011, respectively. Following Groot Zwaaftink et al. (2016), the simulation profiles are averaged for the regions north of 60°N and 170°W to 35°W in April and 135°W to 35°W in July. The pink shade on each panel represents the standard deviation with respect to time and space for the simulated total dust concentrations.

8. Figs 5, 6, 7, 9, 10, 11, 12, 13: Include the time averaging interval in the captions.

**Reply:** Thanks for the suggestion. We have included the time averaging interval for model results in the captions of these figures. Briefly, only Figure 5 is averaged from 2007 to 2009 to match the

period of CALIPSO retrieval, while all the other figures are averaged over the entire simulation period (2007 to 2011).

9. Line 257: I would argue that also the MAM case is consistently underestimated by the model and that DJF is underestimated near the surface. However, this is difficult to judge because neither observations nor model results contain any measure of uncertainty. It may well be that the two actually agree within the level of uncertainty. Please include some discussion about this.

**Reply:** We added year-to-year variability of the simulated dust extinction results in Figure 5. The revised Figure 5 looks as follow.

[Figure]

**Figure 5.** Comparison of seasonal CALIPSO retrieved (Luo et al., 2015a, b) (black solid line) and model simulated (pink solid line; with pink shade representing year-to-year variability) dust extinction vertical profiles in the Arctic. Contributions from seven tagged sources are shown by

colored dashed lines. The CALIPSO retrieval is for the year 2007 to 2009, while the model is averaged over the same years.

We agree with the reviewer on the consistent underestimation in MAM and the near surface underestimation in DJF. We add some discussions regarding these issues in Section 3.1:

Line 274-280: "The simulated dust extinction also shows a consistent underestimation in springtime (MAM) and a near surface underestimation in wintertime (DJF). Since the Arctic is mostly covered by ice and snow in these two seasons, the impacts of HLD are expected to be limited and the low biases are most likely due to the underprediction of LLD transport. The near surface underestimation in DJF may indicate a too weak LLD transport in the lower troposphere (e.g., the transport of dust emitted from Central Asia; see Figure 7 and the corresponding discussions in Section 3.2)."

10. Figure 6: It would be instructive to additionally represent this figure with percentage contributions rather with absolute values for the column burden. This would make it easier to convey the information how much each region contributes to the burden in a given location.

**Reply:** We have added the spatial distribution of percentage contributions from each source to the annual mean dust column burden in the supplementary (new Figure S2). The new figure looks as follow.

[Figure]

**Figure S2.** Global distribution of relative contributions (%) to the annual mean (2007 to 2011) dust column burden from each tagged source region.

We also added/modified some descriptions/discussions about the new Figure in the revised Section 3.2:

Line 287-289: "The transport pathways can be identified from the dust burden spatial distribution for each source in Figure 6, while the relative contribution of each source to the total dust burden is shown in Figure S2."

Line 295-299: "As shown in Figure 6a and Figure S2a, the local dust is confined within the high latitudes, with the higher amounts and higher contributions to the total dust burden near the sources in North Canada, coast of Greenland, and Iceland. The interior of the Greenland ice sheet, with its higher elevations, is more influenced by LLD from North Africa and East Asia than HLD (Figure S2c and S2f)."

Line 320-321: "Overall, the LLD from North Africa and Asia contributes more to the Eurasia and Pacific sector of the Arctic (Figures S2c to S2f)."

> 11. Figure 8: Suggest adding quantity and unit (INP conc. / $L^{-1}$) as axes labels. How are the temperatures chosen – are they determined by what was used in the observations? How exactly were the time intervals of observations and measurements matched up? Line 346 mentions "monthly averaged aerosol populations" while Table 3 only lists Spring/Summer/Autumn of various years.

**Reply:** We have changed the axes labels of Figure 8 to "Mod. INP conc. ($L^{-1}$)" and "Obs. INP conc. ($L^{-1}$)" following your suggestion. As you mentioned, the temperatures we used are determined by what was reported in the INP measurements. We did not intentionally do any average for the observation data we have. However, depending on the data availability, the data we have may already be processed to monthly averages (datasets A, C, E, H in Table 3). Also, some of the datasets (datasets D, G, I) have too many samples to be plotted clearly. To improve the presentation of the figure, we randomly select 15% of the data points in these datasets to plot (we did not take average). We have confirmed that the range of the selected samples is very close to the whole dataset. For each observed data point (i.e., each symbol on Figure 8), we used the model monthly averaged aerosol properties of the corresponding month from year 2007 to 2011 to diagnose the simulated INP concentrations. To clarify this, we added the exact months of the sampling period for each measurement in Table 3, column 3. The new Table 3 looks as follow.

**Table 3.** Summary of the nine Arctic INP measurements used for INP comparisons in Figure 8.

| | Location | Time period | Measured platform | Reference | Possible INP source mentioned in literature | INP source attribution from modeling[+] |
|---|---|---|---|---|---|---|
| A | Utqiaġvik | Apr. 2008 (spring) | Aircraft | McFarquhar et al. (2011) | Metallic or composed of dust[*] | LLD (EAs) |
| B | Alert | Mar. - May 2014 (spring) | Ground-based | Mason et al. (2016) | Not mentioned | LLD (EAs) |
| C | Alert | Mar. 2016 (spring) | Ground-based | Si et al. (2019) | LLD from Gobi Desert | LLD (EAs) |
| D | Zeppelin | Mar. 2017 (spring) | Ground-based | Tobo et al. (2019) | Marine organic aerosols | HLD (NEu) |
| E | Oliktok Point | Mar. - May 2017 (spring) | Ground-based | Creamean et al. (2018) | Dust and primary marine aerosols | LLD (mainly from EAs and some from NAf) |
| F | Alert | Jun. - Jul. 2014 (summer) | Ground-based | Mason et al. (2016) | Not mentioned | HLD (NCa) |
| G | Zeppelin | Jul. 2016 (summer) | Ground-based | Tobo et al. (2019) | HLD from Svalbard or other high latitude sources[**] | HLD (NEu) |
| H | Utqiaġvik | Oct. 2004 (autumn) | Aircraft | Prenni et al. (2007) | Dust and carbonaceous particles | HLD (NCa) and LLD (EAs) |
| I | South of Iceland | Oct. 2014 (autumn) | Aircraft | Sanchez-Marroquin et al. (2020) | Icelandic dust | Dominated by HLD (GrI), little from LLD (NAf) |

[+] The modeling analyses include INP contribution from HLD (using SM20), LLD (using D15), BC, and SSA.
[*] Carbonate, black carbon, and organic may also contribute, according to Hiranuma et al. (2013).
[**] The HLD in this campaign is reported to have remarkably high ice nucleating ability, which may be related to the presence of organic matter.

12. Section 3.3: there are various INP parameterizations out there for various kinds of dust. While I don't expect the authors to compare all of them, I suggest discussing the reason for the choices made for this paper since the choice of the parameterization can have a big impact on the result. (However, I do recommend including the default parameterization, see comment 2 above).

**Reply:** We use D15 for the LLD INP because it produced the most reasonable LLD INP concentrations in EAMv1 based on our earlier study (Shi and Liu, 2019). There are various other LLD INP parameterizations, many of which are based on the ice-nucleating active surface site density (INAS) (e.g., Niemand et al., 2012; Ullrich et al., 2017). The Niemand et al. (2012) parameterization was tested in Shi and Liu (2019), which concluded that Niemand et al. (2012) may have an overestimation of Arctic INP concentrations in EAMv1, which is likely because the simulated aerosol size distribution in the Arctic is biased to the small size range. We expect the other INAS-based schemes to produce a similar results to Niemand et al. (2012). So, we do not use them in our current study. There are also INP parameterizations based on dust minerology (e.g., Atkinson et al., 2013; Harrison et al., 2019), which are not used because we do not represent dust speciation in the current model.

There are not a lot of HLD INP parameterizations (or even data) as compared to LLD INP parameterizations. To our knowledge, Paramonov et al. (2018) analyze the ice nucleation ability of soil samples collected from Iceland and provide an INAS-based fit. We use the SM20 parameterization which was developed based on airborne samples rather than Paramonov et al. (2018) in our study, due to the possible large ice nucleation ability differences between soil samples and airborne dust samples.

Please see our reply to your comment 2 above, we have included the default CNT parameterization in the mode-observation INP comparison.

We add a section in Text S2 in the supporting information to discuss the choice of the dust parameterizations.

**"S2.6 Discussion regarding the choice of dust ice nucleation parameterizations**

In this study, we use three dust ice nucleation parameterizations (i.e., CNT, D15, and SM20). CNT is chosen because it is the default ice nucleation scheme for EAMv1. We use D15 because it is found to produce the most reasonable INP concentrations in EAMv1 based on our earlier study (Shi and Liu, 2019). There are various other LLD INP parameterizations, many of which are INAS-based (e.g., Niemand et al., 2012; Ullrich et al., 2017). The Niemand et al. (2012) parameterization was tested in Shi and Liu (2019) and was found to overestimate the Arctic INP concentrations with corrected dust concentrations in EAMv1. There are also INP parameterizations based on dust minerology (e.g., Atkinson et al., 2013; Harrison et al., 2019), which are not used because we do

not represent dust speciation in the current model. There are not a lot of HLD INP parameterizations (or even data) as compared to LLD INP schemes. To our knowledge, Paramonov et al. (2018) analyze the ice nucleation ability of soil samples collected from Iceland and provide an INAS-based fit. We use SM20 which was developed based on airborne samples rather than Paramonov et al. (2018) in our study, due to the possible large differences between soil samples and airborne dust samples."

13. Code availability: The github link only points to the general E3SM repo. The authors should include the code that was actually used to run the simulations (including the tagging), for example using zenodo archiving: https://guides.github.com/activities/citable-code/

**Reply:** This work is funded by the Department of Energy (DOE) Atmospheric System Research (ASR) Program. Unfortunately, we cannot share the code without the permission from DOE.

Typographical errors:

1. Line 126: Explain WBF

**Reply:** The WBF process has been explained in the revised manuscript:

Line 38-42: "The AMPCs lifetime, properties, and radiative effects are closely connected to the primary ice formation process, as the formed ice crystals grow at the expense of cloud liquid droplets due to the lower saturation vapor pressure with respect to ice than that to liquid water (so-called Wegener-Bergeron-Findeisen process or, in short, WBF process; Liu et al., 2011; M. Zhang et al., 2019)."

2. Line 133: Should read "are shown"

**Reply:** It has been corrected. Thanks.

General comment: Axes labels: I recommend to not use the format 1E-6 etc. to represent numbers, but use $10^{-6}$ etc. instead.

**Reply:** We updated Figs. 1, 2, 3, 6, 8, S1 and S8 following your suggestion.

---

## Author Response (AR2)

We thank the two anonymous reviewers for their encouraging comments. Below, we explain how the comments are addressed and make note of the revisions in the revised manuscript. The reviewers' comments are in blue color. Our replies are in black, and our corresponding revisions in the manuscript are in red.

**Responses to reviewer #1**

I thank the authors for their response and careful consideration of my suggestions. Other than requesting more additional clarification on the MODIS product used which appears to require another product for comparison (please see below), I only have some minor suggestions to make regarding the inclusion of some of the details in the responses to the manuscript itself:

**Reply:** Thanks for your time and the encouraging comments. Please see our responses to your final comments below.

1) Regarding point #8, what version of MODIS is being used? Are the LWPs from Collection 5 or 6? The Platnick et al. 2003 reference is specifically about MODIS on Terra. The model observations should be compared to the MODIS product designed for comparisons with COSP as described by Pincus et al. 2012 which allows for apples-to-apples comparisons and also uses both Terra and Aqua.

**Reply:** Thanks for pointing this out. The MODIS LWP data we used were from Collection 5.1. Following your suggestion, we replaced it with the Pincus product. The overall conclusion regarding the LWP comparisons does not change. We added some descriptions regarding the Pincus product in the revised manuscript:

Line 542-548: "Two MODIS datasets are used, including the standard Collection 6.1 product (Pincus et al., 2012; P12) and Khanal et al. (2020; K20). The P12 product combines MODIS observations from Terra and Aqua and is designed for apples-to-apples comparisons with modelling results from the Cloud Feedback Model Intercomparison Project (CFMIP) Observation Simulator Package (COSP). The standard product has a well-known positive zonal bias that is strongly correlated with the solar zenith angle (SZA)."

The revised Figure 14 and its caption look as follow.

[Figure]

**Figure 14.** a) Annual mean Arctic (60°N to 80°N in this subplot) averaged LWP over ocean for the MODIS observations and the four simulations (2007-2008). Two MODIS datasets are used, including the standard product (Pincus et al., 2012; P12; averaged from 2007 to 2008) and an improved one (Khanal et al., 2020; K20; averaged from 2007 to 2009). The MODIS simulator is used to calculate the simulated LWP. b) - e) Annual mean Arctic (60°N to 90°N in these subplots) averaged b) FSDS, c) FLDS, d) SWCF, and e) LWCF for the CERES observation (2007-2011) and the four simulations (2007-2011).

Minor suggestions:

1) Regarding point #7, although the dominant role of wet despoliation is speculative, I think it is worth mentioning the authors' plausible explanation in the manuscript (understanding that it is beyond the scope of the manuscript to do a detailed analysis on this).

**Reply:** Thanks for the suggestion. We added a brief discussion regarding this issue in the revised manuscript:

Line 336-341: "For example, the wet removal process is expected to have large discrepancies among different models, because of the large uncertainties in the model representation of clouds and precipitation. The different spatial distributions of dust emission due to the use of different emission parameterizations may also contribute to the discrepancies (e.g., North Africa dust in our study contributes slightly less (51.9%) to the global dust emission than the other studies (from 57% to 67%)."

2) Also regarding point #8, please clarify what the bias in LWP is with respect to. Specifically, the Khanal product reduces the correlation between MODIS LWP and its dependence on large SZA near the poles. Please also include a brief description of how the Khanal product differs from the standard MODIS product, i.e. that the new method utilizes two parameters: the solar zenith angle and cloud heterogeneity index, both of which are available for each MODIS pixel.

**Reply:** We clarified this issue in the revised manuscript as follow.

Line 546-549: "The standard product has a well-known positive zonal bias near the poles that is strongly correlated with the solar zenith angle (SZA). The K20 product largely reduces this bias by utilizing the SZA and cloud heterogeneity index in their retrieval algorithm."

3) Regarding point #9 in response : "…CALIPSO, which significantly improves the detection of optically thin dust layer, especially in the Arctic" — improved compared to what? Please also include this information in the manuscript itself in addition to what was included on lines 266-268.

**Reply:** It is compared to the standard CALIOP Level 2 5 km layer products. More details can be found in Luo et al. (2015b) (see their Figure 3). We included this information in the manuscript.

Line 266-269: "This data set has improvements in dust separation from other aerosol types and thin dust layer detection in the Arctic compared to the standard Cloud-Aerosol Lidar with Orthogonal Polarization (CALIOP) Level 2 product (Winker et al., 2013)."

**Responses to reviewer #2**

The authors did a thorough job with addressing my comments, I really like the revised version of the paper and recommend it for publication.

**Reply:** Thanks for your time and the encouraging comments. Please see our responses to your final comments below.

I only have one follow-up comment regarding Figure 5: You nicely added uncertainty bands to the other comparison figures for model results and observations. Is it possible to assign an uncertainty range to the CALIPSO observations as well?

**Reply:** We added the uncertainty bands to the CALIPSO retrievals in the revised manuscript. We assume the uncertainty bands for the CALIPSO retrievals to be 20% following Yang et al. (2022). The revised Figure 5 and the figure caption look as follow.

[Figure]

**Figure 5.** Comparison of seasonal CALIPSO retrieved (Luo et al., 2015a, b; Yang et al., 2022) (black solid line; with gray shade representing uncertainty) and model simulated (pink solid line; with pink shade representing year-to-year variability) dust extinction vertical profiles in the Arctic (above 60°N). Contributions from seven tagged sources are shown by colored dashed lines. The CALIPSO retrievals are for the year 2007 to 2009, while the model results are averaged over the same years. The uncertainties of the CALIPSO retrievals are assumed to be 20% following Yang et al. (2022).

And lastly, it is unfortunate that the new code developments cannot be made available due to DOE restrictions, but accept that this is not something that can be resolved within this review cycle.

**Reply:** We thank the reviewer for the understanding.